# VIDEOSHIELD: REGULATING DIFFUSION-BASED VIDEO GENERATION MODELS VIA WATERMARKING

**Runyi Hu**[1], **Jie Zhang**[2]*, **Yiming Li**[1], **Jiwei Li**[3], **Qing Guo**[2], **Han Qiu**[4], **Tianwei Zhang**[1]

[1]Nanyang Technological University    [2]CFAR and IHPC, A*STAR, Singapore
[3]Zhejiang University    [4]Tsinghua University
{runyi.hu, ym.li, tianwei.zhang}@ntu.edu.sg
{zhang_jie, guo_qing}@cfar.a-star.edu.sg
jiwei_li@zju.edu.cn    qiuhan@tsinghua.edu.cn

## ABSTRACT

Artificial Intelligence Generated Content (AIGC) has advanced significantly, particularly with the development of video generation models such as text-to-video (T2V) models and image-to-video (I2V) models. However, like other AIGC types, video generation requires robust content control. A common approach is to embed watermarks, but most research has focused on images, with limited attention given to videos. Traditional methods, which embed watermarks frame-by-frame in a post-processing manner, often degrade video quality. In this paper, we propose VIDEOSHIELD, a novel watermarking framework specifically designed for popular diffusion-based video generation models. Unlike post-processing methods, VIDEOSHIELD embeds watermarks directly during video generation, eliminating the need for additional training. To ensure video integrity, we introduce a tamper localization feature that can detect changes both temporally (across frames) and spatially (within individual frames). Our method maps watermark bits to template bits, which are then used to generate watermarked noise during the denoising process. Using DDIM Inversion, we can reverse the video to its original watermarked noise, enabling straightforward watermark extraction. Additionally, template bits allow precise detection for potential temporal and spatial modification. Extensive experiments across various video models (both T2V and I2V models) demonstrate that our method effectively extracts watermarks and detects tamper without compromising video quality. Furthermore, we show that this approach is applicable to image generation models, enabling tamper detection in generated images as well. Codes and models are available at https://github.com/hurunyi/VideoShield.

## 1 INTRODUCTION

In the era of AI-Generated Content (AIGC), the generation of images (Saharia et al., 2022), audios (Huang et al., 2023), and videos (Blattmann et al., 2023) has become increasingly accessible. Among these, video generation stands out as one of the most challenging applications. Recently, significant breakthroughs have been made in this field. With the development of diffusion models (Sohl-Dickstein et al., 2015; Ho et al., 2020) and the scaling law (Kaplan et al., 2020), which shows that increasing the model size and dataset volume enhances performance, modern AI models like Sora (Brooks et al., 2024) are now capable of generating high-quality videos with complex motion dynamics and extended temporal spans. However, these advancements raise serious concerns about content control. AI-generated videos are valuable assets used in various industries, from entertainment to education. Yet, they are vulnerable to unauthorized use and tampering. This underscores the need for robust methods to regulate the origin and modification of such content. Implementing these safeguards is crucial to ensure the integrity of creative works and protect against potential misuse.

To address these challenges, common solutions involve watermarking (Jia et al., 2021; Hu et al., 2024a; Yang et al., 2024) and tamper localization (Dong et al., 2022). Watermarking embeds invisible information in digital content for origin verification, while tamper localization identifies altered

---

*The corresponding author

areas within the content. However, to the best of our knowledge, there is no existing method specifically designed for AI-generated videos. The straightforward approach of embedding and extracting watermarks frame by frame, along with tamper localization, presents several challenges. First, applying image watermarking techniques to individual frames degrades overall video quality. Second, existing image tamper localization methods can only detect spatial tampering within single frames and fail to address the temporal one, such as frame order change. This limitation stems from the inability of image-based methods to process frames collectively. Additionally, most current tamper localization methods are passive and rely on extensive training to identify tampered and original areas, which limits their effectiveness across diverse data distributions, and inducing high overhead.

To remedy these limitations, we propose VIDEOSHIELD, a training-free watermarking framework specifically designed for diffusion-based video models, which achieves watermark embedding in the video generation process, and can simultaneously perform watermark extraction and tamper localization. Inspired by recent works (Wen et al., 2023; Yang et al., 2024), we map watermark bits to Gaussian noise, which is then denoised to generate videos. To enable both watermark extraction and tamper localization, we introduce *template bits* derived from the watermark bits, which are used to generate watermarked noise. The template bits correspond one-to-one with each pixel, exhibiting local fragility essential for tamper localization, while the watermark bits have a one-to-many correspondence with pixels, offering greater tolerance to distortion and ensuring robust extraction. Using Denoising Diffusion Implicit Model (DDIM) Inversion (Song et al., 2020), the denoised video is inverted back to its initial watermarked noise for extraction. When a video is tampered with, both the inverted watermarked noise and template bits are disrupted. To localize the tamper, we compare these disrupted template bits with the originals, using two modules for temporal and spatial localization. Additionally, when performing spatial tamper localization, we employ a Hierarchical Spatial-Temporal Refinement (*HSTR*) module to balance accuracy and granularity, enhancing the overall performance.

In conclusion, our key contributions are as follows:

- We emphasize the importance of protecting content authentication and integrity for AI-generated videos. Additionally, we analyze the limitations of directly applying existing image watermarking and tamper localization methods, such as the degradation of video quality caused by embedding watermarks frame by frame via post-processing, and the poor transferability of passive tamper localization methods.

- We propose VIDEOSHIELD, a training-free video watermarking framework designed to embed watermarks during video generation. It simultaneously enables watermark extraction and tamper localization based on the inverted template bits through DDIM Inversion on the generated video.

- Extensive experiments demonstrate that VIDEOSHIELD effectively extracts watermarks and localizes tamper across various video generation models (T2V and I2V), without compromising video quality. Moreover, VIDEOSHIELD can be seamlessly adapted to localize tamper in images generated by T2I models.

## 2 METHODOLOGY

### 2.1 DESIGN PRINCIPLES

In general, we aim to utilize the embedded watermark to facilitate both subsequent watermark extraction and tamper localization. However, traditional watermarking methods often embed a number of watermark bits that are significantly smaller than the total number of pixels in the cover image to ensure fidelity and robustness. This many-to-one relationship between pixels and bits clearly hinders accurate localization.

In order to address the above challenges, we need a template that has a one-to-one, deterministic relationship with the pixels and is fragile enough to be disrupted when the corresponding pixels are tampered with. Instead of introducing a separate template image like EditGuard (Zhang et al., 2023b) which requires extensive training, we aim to incorporate an intermediate template during the process of watermark bits embedding, avoiding any additional overhead. Based on this consideration and inspired by Gaussian Shading (Yang et al., 2024), we use the Gaussian noise as the carrier for the watermark. During the transformation of the watermark bits into Gaussian noise, we introduce

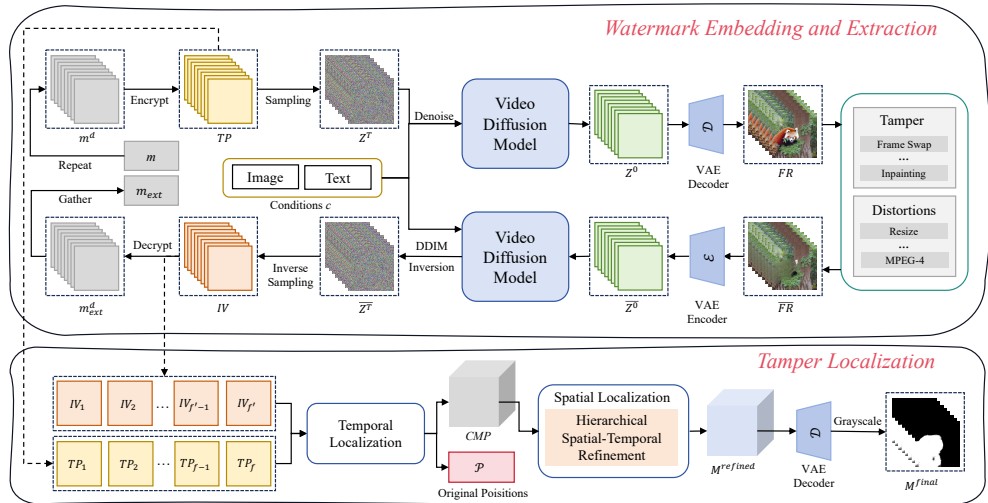

Figure 1: The overall framework of VIDEOSHIELD. (1) In the **Watermark Embedding and Extraction** stage, we first map the watermark bits $m$ to the initial Gaussian noise $Z^T$ via an intermediate set of random template bits $TP$, which are derived from $m$. The video diffusion model $\mathcal{M}$ then iteratively denoises $Z^T$, ultimately generating the video frames $FR$. For watermark extraction, $\mathcal{M}$ uses DDIM Inversion on the tampered or distorted video frames $\overline{FR}$ to recover the inverted noise $\overline{Z^T}$. This noise is then transformed into inverted bits $IV$, from which the watermark bits are extracted. (2) In the **Tamper Localization** stage, $TP$ and $IV$ are processed by a temporal module to localize temporal tamper and restore their temporal positions. The resulting comparison bits matrix $CMP$ is then passed to a spatial module, which incorporates a hierarchical spatio-temporal refinement (*HSTR*) module to enhance localization performance. Both stages are training-free and can be applied to any diffusion-based video generation model.

template bits that can be reversibly converted from the watermark bits. On one hand, the template bits have a one-to-one correspondence with the pixels generated by the diffusion model, which makes them locally fragile to the spatial tamper. On the other hand, the template bits corresponding to each frame of the video are also different, resulting in sensitivity to position. Based on the aforementioned properties of the template bits, we can achieve both temporal and spatial tamper localization. Next, we provide the design details.

## 2.2 OVERVIEW

The framework of VIDEOSHIELD is illustrated in Figure 1. It consists of two key stages: (1) Watermark Embedding and Extraction and (2) Tamper Localization. Overall, VIDEOSHIELD establishes a reversible conversion chain consisting of watermark bits $m$, template bits $TP$, Gaussian noise $Z^T$, and video frames $FR$ to achieve watermark embedding, watermark extraction and tamper localization. The chain can be expressed as $m \Leftrightarrow TP \Leftrightarrow Z^T \Leftrightarrow FR$. Watermark embedding corresponds to the rightward transformation, watermark extraction and tamper localization correspond to the leftward transformation. We briefly explain the rationale behind the choice of each reversible conversion algorithm. The conversion between $m \Leftrightarrow TP$ employs encryption and decryption algorithms, while $TP \Leftrightarrow Z^T$ relies on truncated sampling and inverse sampling. The encryption algorithm ensures that the template bits are completely random, and the noise generated through truncated sampling based on these bits follows a Gaussian distribution. The conversion between $Z^T \Leftrightarrow FR$ involves the denoising and noising processes. For noise addition, we use DDIM sampling, which guarantees the reversibility of the process through the corresponding denoising step, known as DDIM Inversion. Next, we explain the details of each step.

## 2.3 WATERMARK EMBEDDING AND EXTRACTION

Inspired by Gaussian Shading (Yang et al., 2024), we map the watermark bits to Gaussian noise to achieve watermark embedding and use the inverted watermarked noise by DDIM Inversion on the denoised video for watermark extraction.

**Watermark embedding.** We describe how to map the watermark bits to Gaussian noise to achieve watermark embedding. First, we randomly sample watermark bits $m \in \{0, 1\}$ and reshape it to $(\frac{f}{k_f}, \frac{c}{k_c}, \frac{h}{k_h}, \frac{w}{k_w})$, where $f, c, h, w$ are the numbers of frames and channels, height and weight of $Z^T$, $k_*$ is the corresponding repeat factor. Then we repeat $m$ to $m^d$ with the shape of $(f, c, h, w)$ and encrypt $m^d$ to $TP$ with the same shape using ChaCha20 (Bernstein et al., 2008), which is a stream cipher that takes a 256-bit key, a 96-bit nonce, and plaintext as input to produce a pseudo-random ciphertext as output. The algorithm generates a keystream that is XORed with the plaintext for encryption. To get $Z^T$, we use truncated sampling conditioned on $TP$. Assume that the probability density function and percentile function of the Gaussian distribution $\mathcal{N}(0, 1)$ are $f(.)$ and $ppf(.)$ respectively. For every bit $\lambda = TP_{q,i,j,k} \in \{0, 1\}$ in $TP$, the corresponding sampled noise point is $\beta = Z^T_{q,i,j,k}$, where $q, i, j, k$ represent the indices in the $f, c, h, w$ dimensions, respectively. When $\lambda$ is 0, $\beta$ samples from the truncated negative half-interval of $\mathcal{N}(0, 1)$, otherwise, $\beta$ samples from the truncated positive half-interval. The conditional probability distribution followed by $\beta$ can be expressed as:

$$p(\beta | \lambda) = \begin{cases} 2f(\beta), & ppf(\frac{\lambda}{2}) < \beta \le ppf(\frac{\lambda+1}{2}) \\ 0, & otherwise \end{cases}. \tag{1}$$

Then we can determine the probability distribution of $\beta$ as:

$$p(\beta) = \sum_{\lambda=0}^{1} p(\beta|\lambda)p(\lambda) = \frac{1}{2}(p(\beta|0) + p(\beta|1)) = f(\beta). \tag{2}$$

Therefore, $\beta$ follows a standard Gaussian distribution and will not affect the subsequent denoising. Finally, the video diffusion model $\mathcal{M}$ peforms iterative denoising on $Z^T$ to get the denoised latents $Z^0$, which are decoded as $FR$ by the VAE decoder $\mathcal{D}$.

**Watermark extraction.** We explain how to extract watermark bits from the generated video using DDIM Inversion. Before watermark extraction, $FR$ can be tampered with or distorted, resulting in $\overline{FR}$. We compress $\overline{FR}$ into latents $\overline{Z^0}$ through the VAE encoder $\mathcal{E}$, and then use $\mathcal{M}$ to convert $\overline{Z^0}$ into the noised latents $\overline{Z^T}$ via DDIM Inversion. By inverse sampling based on $\overline{Z^T}$, we get the inverted bits $IV$:

$$IV_{q,i,j,k} = \begin{cases} 0, & \overline{Z^T}_{q,i,j,k} \le 0 \\ 1, & \overline{Z^T}_{q,i,j,k} > 0 \end{cases}. \tag{3}$$

Then we decrypt $IV$ to $m^d_{ext}$ for watermark extraction. Since each watermark bit $m_i$ in $m$ is copied $k_{all} = k_f \times k_c \times k_h \times k_w$ times in the watermark repeat stage, $m_i$ actually corresponds to $k_{all}$ bits in $m^d_{ext}$. When more than half of these $k_{all}$ bits are 1, the corresponding extracted bit $m_{ext_i}$ is set to 1; otherwise it is 0. Finally, we retrieve the extracted watermark bits $m_{ext}$ from $m^d_{ext}$.

## 2.4 TAMPER LOCALIZATION

In this section, we provide a detailed explanation of how temporal and spatial tamper localization are achieved by comparing the inverted bits with the template bits.

### 2.4.1 TEMPORAL TAMPER LOCALIZATION

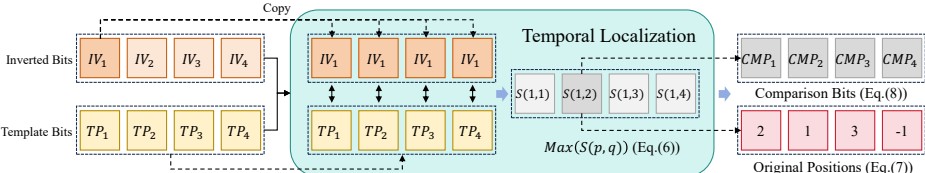

Figure 2: The pipeline of the temporal tamper localization module. We use the first frame ($IV_1$) inside the module (cyan area) as an example to show the localization process.

We focus on temporal tamper localization for the following two scenarios: (1) Temporal tamper, including frame swapping, insertion, and dropping, alters the original sequence of frames, resulting

in semantic changes. (2) Changes in frame order prevent the inverted bits from being compared with the corresponding template bits at their original positions, thereby hindering spatial tamper localization. In general, our core idea is to compare the inverted bits of each frame in the reordered video with the template bits corresponding to all frames in the original video, identifying the position with the highest comparison accuracy to effectively recall its original position for successful localization and restoration. The temporal localization module's pipeline is shown in Figure 2.

Specifically, suppose $p$ and $q$ represent the frame positions in the tampered and original video frames, this module makes $f$ copies of $IV_p$ at every position of $IV$. It then performs a bit-by-bit comparison with $TP$, corresponding to each frame of the original video. This comparison produces the comparison bits $C_{p,q} = [IV_p = TP_q]$, which collectively form the matrix $C$ with a shape of $(f', f, c, h, w)$, where $f'$ indicates the tampered video frames. Then, it calculates the average score across the last three dimensions to obtain the comparison score $S$:

$$S(p,q) = \frac{1}{chw} \sum_{i=1}^{c} \sum_{j=1}^{h} \sum_{k=1}^{w} C_{p,q,i,j,k}. \tag{4}$$

$S(p,q)$ can be used to measure the matching degree between $IV_p$ and $TP_q$. Based on $S(p,q)$, we get the position $M(p)$ in the original video frames which has the highest matching degree by:

$$M(p) = \mathrm{argmax}_{q \in [1,f]} S(p,q). \tag{5}$$

When a tampered frame is not part of the original video (e.g., when a new frame is inserted), there is no corresponding original position. In such cases, we mark these positions as -1 and use $t_{temp}$ to determine whether a frame belongs to the video. As such, we get the original position $\mathcal{P}$ by:

$$\mathcal{P}(p) = \begin{cases} M(p), & S(p, M(p)) > t_{temp} \\ -1, & S(p, M(p)) \le t_{temp} \end{cases}. \tag{6}$$

Finally, the module produces the comparison bits matrix $CMP$ which is used for spatial tamper localization and can be described as:

$$CMP_p = C_{p,M(p)}. \tag{7}$$

### 2.4.2 Spatial tamper Localization

The main process of spatial tamper localization is shown in Figure 3. We first introduce the main process of spatial localization, and then introduce the *HSTR* module in detail.

**Main process.** First, we average the comparison bits matrix $CMP$ with the shape of $(f', c, h, w)$ across the channel dimension to obtain the initial predicted mask $M^{ini}$. The channel average function $CA$ can be expressed as:

$$M^{ini}_{p,j,k} = CA(CMP) = \frac{1}{c} \sum_{i=1}^{c} CMP_{p,i,j,k}. \tag{8}$$

We further use *HSTR* to refine $M^{ini}$ to achieve more accurate localization performance. *HSTR* processes $M^{ini}$ at different levels to get different level masks $\{M^l\}_{l=1,2,...,L}$, where $l$ is level. Then we perform an average on them to get the refined mask: $M^{refined} = \frac{1}{L} \sum_{l=1}^{L} M^l$. Finally, the VAE decoder $\mathcal{D}$ converts $M^{refined}$ to the pixel space, and then applies grayscale conversion to obtain the final extracted mask $M^{final}$.

**Hierarchical Spatial-Temporal Refinement (*HSTR*).** Although $M^{ini}$ provides an initial assessment of watermark at various locations which could indicate potential tampering, its effectiveness is limited. Each location is based on only four comparison bits in the channel dimension, increasing the chance of 'false positives' and reducing accuracy. As shown in Figure 4, increasing the number of comparison bits reduces the variance of comparison accuracy and false positive rates, and enhances the distinction between watermarked and original areas. Thus, *HSTR* aims to refine $M^{ini}$ by using more comparison bits at a high level for accurate judgments. Higher-level improves localization accuracy by gathering more adjacent comparison bits while sacrificing fine granularity, which is in contrast to lower-level. Next, we detail the three parts of *HSTR*.

*Part I: Divide and Gather.* In this part, we explain how to segment the entire video into sub-regions containing different numbers of comparison bits based on different hierarchical levels, and calculate the overall prediction accuracy for each region. Assuming the total hierarchical level is $L$, for every $l \in \{1, 2..., L\}$, we divide $M^{ini}$ into a series of sub-regions with the shape of $(\mu, \mu, \mu)$, where $\mu$ is the local value and $\mu = 2^{l-1}$, indicates the number of adjacent local comparison bits gathered in three dimensions. All the score values contained in each sub-region are gathered and the averaged value is used as the watermark prediction score representing the region. A larger $l$ can increase the number of comparison bits contained in the sub-region, thereby improving the prediction accuracy.

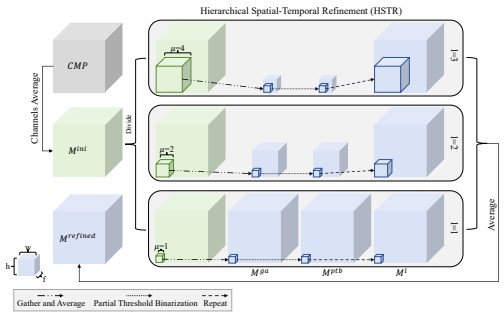

Figure 3: The pipeline of the spatial tamper localization module containing a Hierarchical Spatial-Temporal Refinement (*HSTR*) module. Here, we show the internal workflow of *HSTR* when the total hierarchical level $L = 3$.

Let $b(x) = (x-1)\mu + 1$, the above gather and average function $GA$ to convert $M^{ini}$ with the shape of $(f', h, w)$ to $M^{ga}$ with the shape of $(\frac{f'}{\mu}, \frac{h}{\mu}, \frac{w}{\mu})$ can be described as:

$$M^{ga}_{p,j,k} = GA(M^{ini}_{x,y,z}) = \frac{1}{\mu^3} \sum_{x=b(p)}^{p\mu} \sum_{y=b(j)}^{j\mu} \sum_{z=b(k)}^{k\mu} M^{ini}_{x,y,z}. \tag{9}$$

*Part II: Partial Threshold Binarization.* To further differentiate between tampered and non-tampered areas, we utilize partial threshold binarization $PTB$ for the transformation of $M^{ga}$. We statistically analyze the comparison accuracy distributions $\mathcal{A}_{wm}$ and $\mathcal{A}_{orig}$ on watermarked and original videos corresponding to different levels and find the overlapping area: $\mathcal{Q} = \mathcal{A}_{wm} \cap \mathcal{A}_{orig}$. Suppose the predicted value of an area in $M^{ga}$ is denoted as $o = M^{ga}_{p,j,k}$, when $o \in \mathcal{Q}$, it is impossible to make an effective distinction. When $o \in \mathcal{A}'_{orig} = \mathcal{A}_{orig} \setminus \mathcal{Q}$, it can be concluded with high confidence that the area has been tampered with, so $o$ can be polarized to 0. For $o \in \mathcal{A}'_{wm} = \mathcal{A}_{wm} \setminus \mathcal{Q}$, according to the same principle, $o$ is

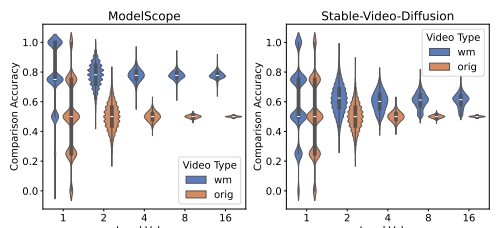

Figure 4: Comparison accuracy distributions of different local values on the watermarked and original videos generated by ModelScope and Stable-Video-Diffusion. Each data point in the figure represents the average accuracy of the sub-region with different local values $\mu$ ($\mu = 2^{l-1}$), containing different numbers of comparison bits.

polarized to 1. Assuming the lower bound of $\mathcal{A}'_{wm}$ and the upper bound of $\mathcal{A}'_{orig}$ are $t_{wm}$ and $t_{orig}$ respectively, $M^{ga}$ can be converted to $M^{ptb}$ as:

$$M^{ptb}_{p,j,k} = PTB(M^{ga}_{p,j,k}) = \begin{cases} 0, & \text{if } o < t_{wm} \\ 1, & \text{if } o > t_{orig} \\ o, & \text{otherwise} \end{cases} \tag{10}$$

For practical usage, in order to determine $t_{wm}$ and $t_{orig}$, we select values corresponding to specific quantile $k$ of $\mathcal{A}_{wm}$ and $\mathcal{A}_{orig}$ to represent the distributions: $[Q_{1-k}(\mathcal{A}), Q_k(\mathcal{A})]$. This is because points with extremely high or low accuracy in the distribution are rare and can be considered outliers, which do not accurately represent the overall distribution, as shown in Figure 4. Therefore, these points can be discarded. A detailed analysis can be found in Sec. 3.4.

*Part III: Repeat.* Finally, we assign the prediction score of each sub-region to all positions within that region to assess the tamper status at different locations. In order to convert the shape of $M^{ptb}$ to $(f', h, w)$, each comparison bit in $M^{ptb}$ is repeated $\mu^3$ times and then filled into the corresponding original sub-region to obtain $M^l$. This operation can be expressed as $R$:

$$M^l_{p,j,k} = R(M^{ptb}) = M^{ptb}_{\lceil \frac{p}{\mu} \rceil, \lceil \frac{j}{\mu} \rceil, \lceil \frac{k}{\mu} \rceil}. \tag{11}$$

Table 1: Comparison with other baseline watermarking methods on videos generated by ModelScope and Stable-Video-Diffusion, respectively. The video quality reported here is the average of all metric values. The **best** results are highlighted in bold.

| Method | ModelScope | | | Stable-Video-Diffusion | | |
|---|---|---|---|---|---|---|
| | Bit Length ↑ | Video Quality ↑ | Bit Accuracy ↑ | Bit Length ↑ | Video Quality ↑ | Bit Accuracy ↑ |
| RivaGAN | 32 | 0.804 | 0.994 | 32 | **0.836** | 0.989 |
| MBRS | 256 | 0.803 | **1.000** | 256 | 0.828 | 0.999 |
| CIN | 30 | 0.756 | **1.000** | 30 | 0.795 | **1.000** |
| PIMoG | 30 | 0.753 | **1.000** | 30 | 0.794 | 0.999 |
| SepMark | 128 | 0.799 | 0.999 | 128 | 0.819 | 0.998 |
| VIDEOSHIELD | **512** | **0.806** | **1.000** | **512** | **0.836** | 0.999 |

## 3 EXPERIMENTS

### 3.1 EXPERIMENTAL SETTING

**Implementation details.** We select two popular open-source models as the default test models: the text-to-video (T2V) model ModelScope (MS) (Wang et al., 2023) and the image-to-video (I2V) model Stable-Video-Diffusion (SVD) (Blattmann et al., 2023). Videos are generated with 16 frames in FP16 mode for both models. The resolutions of the videos generated by the MS and SVD models are 256 and 512, respectively. We use the default sampler and text (image) guidance, with 25 inference steps and 25 inversion steps for both models. A total of 512 watermark bits are embedded into the generated videos. To achieve this, we set $k_f, k_c, k_h, k_w$ to 8, 1, 4, 4 for MS and 8, 1, 8, 8 for SVD. For MS, $k$ in $PTB$ is set to 99, while it is set to 98 for SVD. For both models, $t_{temp}$ and $L$ are set to 0.55 and 3, respectively. For temporal tamper, we evaluate Frame Drop, Frame Insert, and Frame Swap. For spatial tamper, we consider: Crop&Drop (default), STTN (Zeng et al., 2020), and ProPainter (Zhou et al., 2023).

The baseline watermarking methods for comparison are: RivaGAN (Zhang et al., 2019), MBRS (Jia et al., 2021), CIN (Ma et al., 2022), PIMoG (Fang & et al., 2022), and SepMark (Wu et al., 2023). For baseline spatial tamper localization, we compare: MVSS-Net (Dong et al., 2022), OSN (Wu et al., 2022), PSCC-Net (Liu et al., 2022), and HiFi-Net (Guo et al., 2023b). Except for RivaGAN, all methods are open-source and designed for images, as there is limited open-source research in the video domain. When applied to video, the same watermark is embedded in each frame, and tamper localization is performed frame by frame. More implementation details are in Appendix D.

**Datasets.** For MS, we choose 50 prompts from the VBench (Huang et al., 2024) test set, covering five categories: Animal, Human, Plant, Scenery, and Vehicles, with 10 prompts per category. For each prompt, we generate 4 videos, resulting in a total of $50 \times 4 = 200$ videos for evaluation. For SVD, we first employ a text-to-image (T2I) model, specifically Stable Diffusion 2.1 (AI, 2022), to generate 200 images corresponding to the 200 prompts used in the MS evaluation. These images are then used to create 200 videos for evaluation. Additionally, we gather 5 real images from each of the 5 categories, generating a total of $5 \times 5 \times 4 = 100$ videos for evaluation. Except for evaluating spatial tamper localization with STTN and ProPainter, where we use 1/5 of the generated videos for manual annotation (as detailed in Appendix D.3), we use all the generated videos in other cases by default. To statistically analyze $\mathcal{A}_{wm}$ and $\mathcal{A}_{orig}$ to obtain $t_{wm}$ and $t_{orig}$, we further generate 100 watermarked videos and 100 original videos that are not included in the aforementioned dataset for both MS and SVD. More details about our constructed datasets can be found in Appendix C.

**Metric.** For watermark extraction, we use Bit Accuracy which indicates the ratio of correctly extracted bits. To evaluate the quality of the watermarked videos, we adopt five metrics from VBench (Huang et al., 2024): Subject Consistency, Background Consistency, Motion Smoothness, Dynamic Degree, and Imaging Quality. For temporal localization, we measure accuracy using the formula: Accuracy $= \frac{1}{N} \sum_{i=1}^{N} \mathcal{P}_i = \mathcal{O}_i$, where $\mathcal{P}_i$ and $\mathcal{O}_i$ are the predicted position and original position of $frame_i$, and $N$ is the total number of test frames. For spatial localization, following Dong et al. (2022); Liu et al. (2022), we evaluate using F1, Precision, Recall, AUC, and IoU metrics computed between the final extracted mask $M^{final}$ and the ground truth mask $M^{gt}$. When specific metrics are not reported, we default to presenting the average values of these metrics.

### 3.2 MAIN RESULTS

Table 3: Baseline comparison results of spatial tamper localization on different tamper types. All evaluation metrics are better when higher. * represents an outlier and has no practical meaning. Because we find that HiFi-Net correspondence predicts every frame of the video as tampered, so the Recall value is higher.

| Method | STTN | | | | | | ProPainter | | | | | | Crop&Drop | | | | | |
|---|---|---|---|---|---|---|---|---|---|---|---|---|---|---|---|---|---|---|
| | F1 | Precision | Recall | AUC | IoU | Average | F1 | Precision | Recall | AUC | IoU | Average | F1 | Precision | Recall | AUC | IoU | Average |
| ModelScope | | | | | | | | | | | | | | | | | | |
| MVSS-Net | 0.125 | 0.333 | 0.106 | 0.550 | 0.102 | 0.243 | 0.012 | 0.114 | 0.008 | 0.502 | 0.008 | 0.129 | 0.291 | 0.350 | 0.296 | 0.474 | 0.248 | 0.332 |
| OSN | 0.137 | 0.390 | 0.120 | 0.539 | 0.108 | 0.259 | 0.069 | 0.153 | 0.078 | 0.506 | 0.047 | 0.171 | 0.321 | 0.383 | 0.365 | 0.522 | 0.258 | 0.370 |
| PSCC-Net | 0.521 | 0.504 | 0.805 | 0.588 | 0.385 | 0.561 | 0.442 | 0.415 | 0.730 | 0.515 | 0.320 | 0.484 | 0.529 | 0.499 | 0.603 | 0.539 | 0.477 | 0.529 |
| HiFi-Net | 0.003 | 0.211 | 0.001 | 0.497 | 0.001 | 0.143 | 0.532 | 0.391 | 0.999* | 0.500 | 0.391 | 0.563 | 0.423 | 0.308 | 0.975* | 0.491 | 0.306 | 0.501 |
| VIDEOSHIELD | **0.893** | **0.918** | **0.888** | **0.911** | **0.822** | **0.886** | **0.909** | **0.910** | **0.917** | **0.921** | **0.841** | **0.900** | **0.911** | **0.888** | **0.942** | **0.946** | **0.841** | **0.906** |
| Stable-Video-Diffusion | | | | | | | | | | | | | | | | | | |
| MVSS-Net | 0.126 | 0.417 | 0.101 | 0.540 | 0.096 | 0.256 | 0.041 | 0.172 | 0.030 | 0.505 | 0.027 | 0.155 | 0.460 | 0.464 | 0.490 | 0.566 | 0.401 | 0.476 |
| OSN | 0.213 | 0.278 | 0.234 | 0.542 | 0.155 | 0.284 | 0.152 | 0.244 | 0.173 | 0.498 | 0.101 | 0.234 | 0.351 | 0.396 | 0.437 | 0.572 | 0.252 | 0.402 |
| PSCC-Net | 0.405 | 0.377 | 0.732 | 0.526 | 0.290 | 0.466 | 0.385 | 0.318 | 0.757 | 0.496 | 0.272 | 0.446 | 0.501 | 0.409 | 0.738 | 0.571 | 0.372 | 0.518 |
| HiFi-Net | 0.000 | 0.126 | 0.000 | 0.499 | 0.000 | 0.125 | 0.365 | 0.344 | 0.492 | 0.515 | 0.234 | 0.390 | 0.221 | 0.325 | 0.224 | 0.485 | 0.133 | 0.278 |
| VIDEOSHIELD | **0.759** | **0.772** | **0.820** | **0.824** | **0.643** | **0.764** | **0.764** | **0.768** | **0.829** | **0.828** | **0.649** | **0.767** | **0.743** | **0.701** | **0.888** | **0.843** | **0.632** | **0.761** |

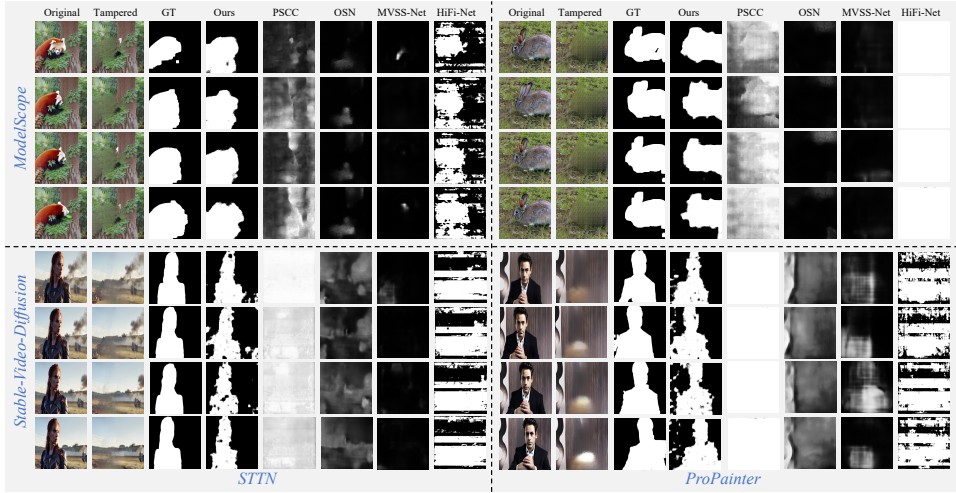

Figure 5: Some visual examples comparison of different spatial localization methods.

**Watermark extraction.** As shown in Table 1, compared to the baseline methods, VIDEOSHIELD offers several significant advantages. First, VIDEOSHIELD embeds substantially more bits which are essential for high-capacity applications. Additionally, VIDEOSHIELD achieves comparable Bit Accuracy to the other methods, proving its effectiveness. Furthermore, the watermarked videos generated by VIDEOSHIELD exhibit noticeably higher video quality because it does not require any post-processing which often degrades video quality.

**Tamper localization.** The temporal tamper localization results of VIDEOSHIELD are presented in Table 2. Note that we are the first to localize temporal tamper in videos, so there is no baseline to compare with. We observe that VIDEOSHIELD demonstrates effective localization capabilities for three different types of temporal tamper. For spatial tamper localization, we compare VIDEOSHIELD against four baseline methods, with the results presented in Table 3. VIDEOSHIELD demonstrates effectiveness across three distinct forms of spatial tamper. In contrast, the baseline methods struggle to localize tamper in the generated videos, underscoring their lack of generalization when applied to videos or other datasets with distributions different from those used during training. Visual examples of tamper localization for different methods are shown in Figure 5.

Table 2: Results of temporal tamper localization.

| Model | Accuracy↑ | | | |
|---|---|---|---|---|
| | Frame Swap | Frame Insert | Frame Drop | Average |
| MS | 1.000 | 1.000 | 1.000 | 1.000 |
| SVD | 0.935 | 0.937 | 0.936 | 0.936 |

The results of VIDEOSHIELD on videos generated by SVD on real conditional images are shown in Table 4, which further confirms the effectiveness of our method.

Table 4: Performance on SVD's generated videos conditioned on different types of images.

| Conditional Image | Video Quality | Bit Accuracy | Spatial Localization | | | Temporal Localization | | |
|---|---|---|---|---|---|---|---|---|
| | | | STTN | ProPainter | Crop&Drop | Frame Swap | Frame Insert | Frame Drop |
| Generated | 0.836 | 0.999 | 0.764 | 0.767 | 0.761 | 0.935 | 0.937 | 0.936 |
| Real | 0.865 | 0.999 | 0.785 | 0.790 | 0.755 | 0.962 | 0.971 | 0.965 |

## 3.3 ABLATION STUDY

**Importance of $PTB$ in *HSTR* and impact of the quantile $k$ in $PTB$.** As shown in Table 5, when $PTB$ is not used, the localization performance is greatly reduced, confirming $PTB$'s importance. In addition, consistent with our previous analysis, the effect is better when $k$ is not 100, that is, when the entire distribution is not used to obtain $t_{wm}$ and $t_{orig}$. Specifically, when $k$ is set to 99 for ModelScope and 98 for Stable-Video-Diffusion, the performance is best, which is also our default setting. Please refer to Appendix F for the ranges of $\mathcal{A}_{wm}$ and $\mathcal{A}_{orig}$ obtained with different $k$ values, along with a more detailed analysis.

**Impact of the total hierarchical level $L$ of *HSTR*.** We explore the optimal hierarchical level $L$ and delve into the importance of *HSTR* module. As shown in Figure 6, when $L$ is set to 3, both ModelScope and Stable-Video-Diffusion achieve the highest average across all metrics, indicating optimal performance in spatial tamper localization. Additionally, we observe that increasing $L$ consistently enhances Precision; however, once it surpasses a certain threshold, Recall begins to decline (3 for Mod-

Table 5: Spatial tamper localization performance under different $k$ in $PTB$. None means without the $PTB$ module.

| Model | $k$ | | | | |
|---|---|---|---|---|---|
| | None | 97 | 98 | 99 | 100 |
| MS | 0.857 | 0.902 | 0.902 | **0.906** | 0.887 |
| SVD | 0.689 | 0.757 | **0.761** | 0.756 | 0.714 |

elScope and 2 for Stable-Video-Diffusion). We posit that when the level exceeds this threshold, it heightens the accuracy of judgment for the entire area. Consequently, all positions within this area are uniformly classified, which reduces the granularity of the judgment and adversely impacts Recall. The visual spatial tamper localization results of different $L$ are provided in Appendix G.1 (Figure 14).

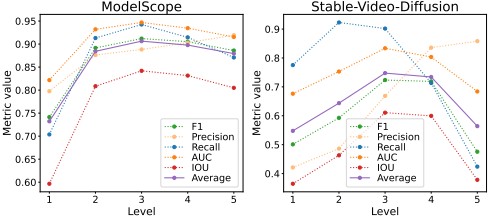
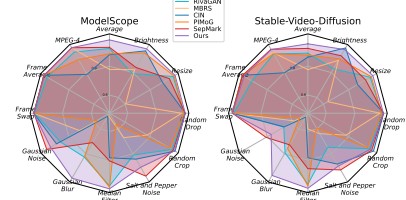

Figure 6: The line chart depicting the variation of different metrics of spatial tamper localizations as hierarchical level $L$ changes.

Figure 7: Comparison of watermark extraction accuracy across different methods under various distortions.

**Impact of different inference and inversion configurations.** Through our experiments, we find that VIDEOSHIELD achieves satisfactory results in both watermark extraction and tamper localization across various inference and inversion settings, with detailed results available in Appendix E. For Stable-Video-Diffusion, increasing the inference steps and reducing image guidance enhance the performance of tamper localization. This improvement likely arises from these settings enabling the model to generate higher-quality dynamic videos, which facilitates better integration of the corresponding template bits into the generated videos for effective tamper localization. For a comprehensive analysis and visualization of the results, please refer to Appendix G.3.

## 3.4 GENERALITY

**Results on other video models.** We test VIDEOSHIELD's effectiveness on two other popular video models: the T2V model ZeroScope (ZS) (Face & cerspense, 2023), and the I2V model I2VGen-XL (I2VGen) (Zhang et al., 2023a). The implementation details can be found in Appendix D.4 and the results are shown in Table 6. It can be seen that VIDEOSHIELD can also achieve effective watermark extraction and tamper localization on both models. However, regarding the tamper localization, VIDEOSHIELD with Stable-Video-Diffusion exhibits a certain gap in performance compared with the other three models. We find that this is due to the lower quality of the generated videos (see Appendix G.3), which leads to the watermark bits not integrating well with the generated videos to achieve better tamper localization. More detailed analysis and visual results of this section can be found in Appendix G.2.

Table 6: Results on other video models beyond MS and SVD in terms of three test dimensions.

| Dimension | MS | SVD | ZS | I2VGen |
|---|---|---|---|---|
| Watermark | 1.000 | 0.999 | 1.000 | 0.999 |
| Temporal | 1.000 | 0.936 | 1.000 | 0.983 |
| Spatial | 0.906 | 0.761 | 0.900 | 0.867 |

Table 7: Tamper localization results on different versions of Stable Diffusion.

| Version | F1 | Precision | Recall | AUC | IoU | Average |
|---|---|---|---|---|---|---|
| 1.4 | 0.947 | 0.921 | 0.976 | 0.947 | 0.901 | 0.938 |
| 1.5 | 0.947 | 0.921 | 0.977 | 0.947 | 0.902 | 0.939 |
| 2.1 | 0.938 | 0.946 | 0.934 | 0.935 | 0.887 | 0.928 |

Table 8: Spatial tamper localization performance under different spatial distortions. In this context, 'w/o' refers to using $t_{wm}$ and $t_{orig}$ derived from the clean watermarked videos, while 'w/' indicates that various distortions are introduced during the distribution testing, leading to adjustments in $t_{wm}$ and $t_{orig}$.

| Model | Adjustment | MPEG-4 | Frame Average | G Noise | G Blur | Median Blur | S&P Noise | Resize | Brightness | Average | Clean |
|---|---|---|---|---|---|---|---|---|---|---|---|
| MS | w/o | 0.838 | 0.641 | 0.559 | 0.644 | 0.737 | 0.537 | 0.797 | 0.622 | 0.672 | 0.906 |
|  | w | 0.801 | 0.774 | 0.641 | 0.787 | 0.799 | 0.610 | 0.809 | 0.701 | 0.740 | 0.806 |
| SVD | w/o | 0.659 | 0.614 | 0.521 | 0.587 | 0.683 | 0.551 | 0.681 | 0.630 | 0.616 | 0.761 |
|  | w | 0.675 | 0.731 | 0.547 | 0.686 | 0.716 | 0.604 | 0.713 | 0.676 | 0.669 | 0.708 |

**Spatial tamper localization on images generated by T2I models.** VIDEOSHIELD can be flexibly adapted to T2I models for the purpose of image spatial tamper localization. We test it on three different versions of the Stable Diffusion model: 1.4, 1.5, and 2.1. The implementation details can be found in Appendix D.4 and the specific test results are presented in Table 7. It is observed that VIDEOSHIELD achieves high performance in tamper localization, further validating the effectiveness of our approach. We provide some visual localization results in Appendix G.2.

### 3.5 ROBUSTNESS

In this section, we further test VIDEOSHIELD's performance of watermark extraction and spatial tamper localization in the face of different distortions to verify its robustness, and the results are shown in Figure 7 and Table 8, respectively. For detailed configurations of the distortions, please refer to Appendix D.5.

**Watermark extraction.** As illustrated in Figure 7, our method demonstrates significantly greater robustness compared to all baseline approaches across almost all types of distortions, achieving an average accuracy of 0.983 on MS and 0.955 on SVD, respectively.

**Spatial tamper localization.** Our tests show that the performance of spatial tamper localization decreases when tampered videos are subjected to further distortions. We attribute this decline to two main factors. First, distortion can be seen as a type of tamper, causing more regions to be classified as tampered and reducing precision. Second, distortions affect $\mathcal{A}_{wm}$ in $PTB$, making the threshold calculated from clean watermarked videos less effective for distinguishing tampered regions in distorted videos. To address this, we introduce random distortions when testing $\mathcal{A}_{wm}$, which allows us to adjust $t_{wm}$ and $t_{orig}$ accordingly. As shown in Table 8, while adjusting the threshold reduces localization performance for clean tampered videos, it significantly improves robustness against various distortions. We also find that setting $L$ as 4 instead of 3 for SVD enhances robustness, as discussed in Appendix F. This demonstrates the flexibility of our method, allowing for multiple thresholds and $L$ values in practical tamper localization applications.

## 4 CONCLUSION

In this paper, we propose VIDEOSHIELD, a training-free video watermarking framework that embeds watermarks during video generation, and achieves watermark extraction and tamper localization during detection. We map the watermark bits to watermarked Gaussian noise for sampling and innovatively introduce template bits during the mapping process. Through DDIM Inversion, the generated video can be converted into template bits, which are then used for both watermark extraction and tamper localization. Extensive experiments demonstrate the effectiveness of VIDEOSHIELD in both tasks and its adaptability across various video and image generation models.

### ACKNOWLEDGMENTS

This research / project is supported by the National Research Foundation, Singapore and Infocomm Media Development Authority under its Trust Tech Funding Initiative (No. DTCRGC-04). It is

also supported by the National Research Foundation, Singapore, and Cyber Security Agency of Singapore under its National Cybersecurity R&D Programme and CyberSG R&D Cyber Research Programme Office. Any opinions, findings and conclusions or recommendations expressed in this material are those of the author(s) and do not reflect the views of National Research Foundation, Singapore, Infocomm Media Development Authority, Cyber Security Agency of Singapore as well as CyberSG R&D Programme Office, Singapore.

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

## A  BACKGROUND

### A.1  DIFFUSION-BASED VIDEO GENERATION MODELS

Diffusion Models (DMs) were first introduced in (Sohl-Dickstein et al., 2015), where the original data is gradually diffused into noise, and the model learns the reverse process. Denoising Diffusion Probabilistic Models (DDPMs(Ho et al., 2020)) improved DMs by framing the reverse process as denoising, streamlining the training and generation phases. Denoising Diffusion Implicit Models (DDIMs (Song et al., 2020)) further enhanced DDPMs, offering faster sample generation with high quality. Due to the deterministic and non-Markovian process, DDIMs allow for nearly reversible noising and denoising. Leveraging DDIM Inversion for image reconstruction is a key feature of our method. To reduce memory usage and speed up training and inference, Latent Diffusion Models (LDMs (Rombach et al., 2022)) use a Variational Autoencoder (VAE) to compress data into a latent space for noise processing, driving the advancement of diffusion-based models.

Recent popular video generation models are mostly built on LDMs, with Sora (Brooks et al., 2024) marking a major shift in the field. Before Sora, models like ModelScope (Wang et al., 2023) and Stable-Video-Diffusion (Blattmann et al., 2023), compress each frame of a video into the latent space using 2D VAEs and employ a U-Net architecture to learn denoising across both spatial and temporal dimensions. However, this approach leads to videos with poor temporal consistency, limited motion range, and shorter durations, while it is also difficult to scale the U-Net to achieve better performance. After Sora, models like Vidu (Bao et al., 2024) and KLING (Kuaishou Large Model Team, 2024) adopt 3D VAEs, which compress the entire video into a series of 3D spacetime patches. This enables the model to naturally capture both spatial and temporal relationships. These models use the more scalable Diffusion Transformer (DiT) (Peebles & Xie, 2023) architecture, allowing for the generation of longer, higher-quality videos with smoother motions and better temporal consistency.

### A.2  WATERMARKING

Watermarking methods[1] for AIGC can be divided into two main categories: *post-processing* and *in-generation*. Post-processing watermarking typically uses an encoder-noise layer-decoder training architecture. The encoder embeds invisible watermarks and the decoder extracts them after distortions are added to the noise layer. Various works introduce specific distortions for different types of robustness: MBRS (Jia et al., 2021) simulates JPEG artifacts, PIMoG (Fang & et al., 2022) targets physical distortions, SepMark (Wu et al., 2023) addresses Deepfake-induced distortions, and Robust-Wide (Hu et al., 2024b) handles semantic-level changes. While most post-processing methods focus on images, RivaGAN (Zhang et al., 2019) and DVMark (Luo et al., 2023) extend them to videos by using 3D convolutions. In-generation watermarking, which embeds watermarks during content creation, is the focus of this paper and is primarily designed for diffusion models. Tree-Ring (Wen et al., 2023) and Gaussian Shading (Yang et al., 2024) can be classified into this category and show strong robustness. Specifically, Tree-Ring embeds multi-ring shaped watermarks in the initial Gaussian noise to achieve 0-bit watermarking, while Gaussian Shading (Yang et al., 2024) embeds multi-bits into the noise. They both adopt DDIM Inversion to acquire the watermarked

---

[1]Note that watermarking techniques can also be used to protect the copyright of data (Guo et al., 2023a; 2024; Li et al., 2025a) and models (Gan et al., 2023; Li et al., 2025b; Shao et al., 2025) other than solely AIGC. However, these topics are out of the scope of this paper.

noise for watermark extraction. However, to our best knowledge, there is currently no in-generation watermarking designed for video generation models.

### A.3    TAMPER LOCALIZATION

Current tamper localization methods can be divided into two categories: *proactive detection* and *passive detection*. Most existing methods fall under passive detection, such as MVSS-Net (Dong et al., 2022) and PSCC-Net (Liu et al., 2022), which learn the boundary features between tampered and original areas by training on tampered data. However, these methods often exhibit poor generalization, making it challenging to transfer effectively across datasets with different distributions. Proactive detection is an alternative approach to tamper localization, which involves embedding additional information into the content to enable tamper localization and is adopted in our work. Recently proposed methods like EditGuard (Zhang et al., 2023b) and V2A-Mark (Zhang et al., 2024) fall within this category and share similar functionalities with our approach. Their core concept involves embedding both watermark bits and template images into images or videos by post-processing, enabling simultaneous copyright detection and tamper localization. However, these methods require extensive training, and their post-processing nature can lead to resource consumption and degradation of image or video quality. Unfortunately, we cannot use these methods as baselines for comparison due to the lack of open-source access to their models.

## B    RELATIONSHIP WITH GAUSSIAN SHADING

VIDEOSHIELD is not a straightforward extension of Gaussian Shading (GS) (Yang et al., 2024) to the video modality. The distinct contributions of our work are outlined as follows:

**1. The Core Role of Template Bits.** Template bits form the cornerstone of our framework, a concept entirely absent in GS. These bits establish a seamless, training-free connection between watermark embedding, extraction, and tamper localization. Furthermore, template bits are highly versatile, applicable to both images and videos, and adaptable to other modalities. This plug-and-play capability paves the way for multifunctional watermarking solutions, making the framework flexible and widely applicable.

**2. GS as One Implementation Method Among Others.** While GS provides one method to implement template bits, it is not the only option. For example, PRC (Gunn et al., 2024) also serves as an alternative implementation. Specifically, PRC employs pseudorandom error-correcting code (PRC) to transform watermark bits into pseudo-random bits that correspond one-to-one with the content. These pseudo-random bits can then be used as template bits. PRC further combines the absolute values of randomly sampled Gaussian noise with the signs of the template bits, embedding the watermark effectively.

**3. Innovations in the Video Modality.** Our work introduces the novel concept of temporal tamper localization, addressing challenges that traditional methods cannot resolve. Additionally, leveraging the unique characteristics of video data, we propose two innovative techniques, **Hierarchical Spatial-Temporal Refinement** (*HSTR*) and **Partial Threshold Binarization** (*PTB*), which substantially enhance the robustness of spatial tamper localization. These contributions close critical gaps in tamper localization for video modalities, setting a foundation for more robust and versatile watermarking frameworks.

## C    DATASETS

### C.1    PROMPTS USED FOR T2V MODELS

The five categories of prompts we used for video generation are as follows.

**Animal**

"a red panda eating leaves"

"a squirrel eating nuts"

"a cute pomeranian dog playing with a soccer ball"

"curious cat sitting and looking around"

"wild rabbit in a green meadow"

"underwater footage of an octopus in a coral reef"

"hedgehog crossing road in forest"

"shark swimming in the sea"

"an african penguin walking on a beach"

"a tortoise covered with algae"

**Human**

"a boy covering a rose flower with a dome glass"

"boy sitting on grass petting a dog"

"a child playing with water"

"couple dancing slow dance with sun glare"

"elderly man lifting kettlebell"

"young dancer practicing at home"

"a man in a hoodie and woman with a red bandana talking to "each other and smiling""

"a woman fighter in her cosplay costume"

"a happy kid playing the ukulele"

"a person walking on a wet wooden bridge"

**Plant**

"plant with blooming flowers"

"close up view of a white christmas tree"

"dropping flower petals on a wooden bowl"

"a close up shot of gypsophila flower"

"a stack of dried leaves burning in a forest"

"drone footage of a tree on farm field"

"shot of a palm tree swaying with the wind"

"candle wax dripping on flower petals"

"forest trees and a medieval castle at sunset"

"a mossy fountain and green plants in a botanical garden"

**Scenery**

"scenery of a relaxing beach"

"fireworks display in the sky at night"

"waterfalls in between mountain"

"exotic view of a riverfront city"

"scenic video of sunset"

"view of houses with bush fence under a blue and cloudy sky"

"boat sailing in the ocean"

"view of golden domed church"

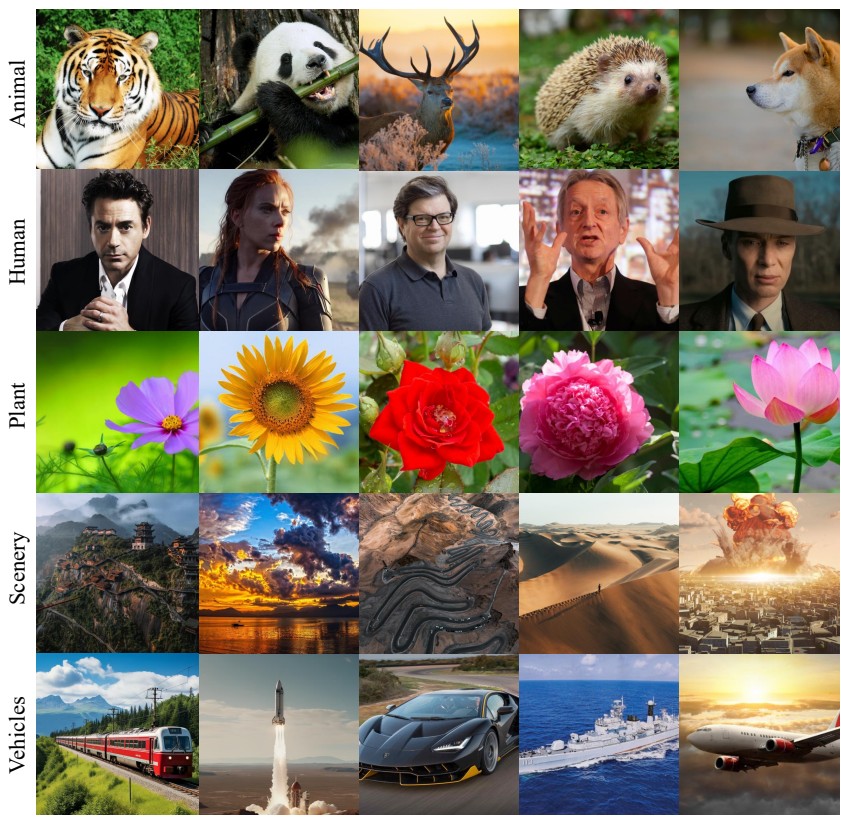

Figure 8: Real images of five categories used for Image-to-Video models.

"a blooming cherry blossom tree under a blue sky with white clouds"

"aerial view of a palace"

**Vehicles**

"a helicopter flying under blue sky"

"red vehicle driving on field"

"aerial view of a train passing by a bridge"

"red bus in a rainy city"

"an airplane in the sky"

"helicopter landing on the street"

"boat sailing in the middle of the ocean"

"video of a kayak boat in a river"

"traffic on busy city street"

"slow motion footage of a racing car"

## C.2 Real images used for I2V models

We present 25 real images used by the I2V model in Figure 8.

## D  MORE IMPLEMENTATION DETAILS

### D.1  VIDEO QUALITY METRICS CALCULATION DETAILS

We present the formulas for calculating the five video quality metrics from VBench (Huang et al., 2024)—Subject Consistency, Background Consistency, Motion Smoothness, Dynamic Degree, and Imaging Quality. Further details are available in the original paper.

**Subject Consistency.**  Subject Consistency is acquired by calculating the DINO (Caron et al., 2021) feature similarity across frames:

$$S_{\text{subject}} = \frac{1}{T-1} \sum_{t=2}^{T} \frac{1}{2} \left( \langle d_1 \cdot d_t \rangle + \langle d_{t-1} \cdot d_t \rangle \right),$$

(12)

where $d_i$ is the DINO image feature of the $i^{th}$ frame, normalized to unit length, and $\langle \cdot \rangle$ is the dot product operation for calculating cosine similarity.

**Background Consistency.**  Background Consistency evaluates the temporal consistency of the background scenes by calculating CLIP (Radford et al., 2021) feature similarity across frames:

$$S_{\text{background}} = \frac{1}{T-1} \sum_{t=2}^{T} \frac{1}{2} \left( \langle c_1 \cdot c_t \rangle + \langle c_{t-1} \cdot c_t \rangle \right),$$

(13)

where $c_i$ represents the CLIP image feature of the $i^{th}$ frame, normalized to unit length.

**Motion Smoothness.**  Motion Smoothness is evaluated by the frame-by-frame motion prior to video frame interpolation models (Li et al., 2023). Specifically, given a generated video consisting of frames $[f_0, f_1, f_2, f_3, f_4, ..., f_{2n-2}, f_{2n-1}, f_{2n}]$, the odd-number frames are manually dropped to obtain a lower-frame-rate video $[f_0, f_2, f_4, ..., f_{2n-2}, f_{2n}]$, and video frame interpolation (Li et al., 2023) is used to infer the dropped frames $[\hat{f}_1, \hat{f}_3, ..., \hat{f}_{2n-1}]$. Then the Mean Absolute Error (MAE) between the reconstructed frames and the original dropped frames is computed and normalized into $[0, 1]$, with a larger number implying smoother motion.

**Dynamic Degree.**  Dynamic Degree is designed to assess the extent to which models tend to generate non-static videos. RAFT (Teed & Deng, 2020) is used to estimate optical flow strengths between consecutive frames of a generated video. Then the average of the largest 5% optical flows (considering the movement of small objects in the video) is taken as the basis to determine whether the video is static. The final dynamic degree score is calculated by measuring the proportion of non-static videos generated by the model.

**Imaging Quality.**  Imaging Quality is measured by the MUSIQ (Ke et al., 2021) image quality predictor trained on the SPAQ (Fang et al., 2020) dataset, which is capable of handling variable-sized aspect ratios and resolutions. The frame-wise score is linearly normalized to [0, 1] by dividing 100, and the final score is then calculated by averaging the frame-wise scores across the entire video sequence.

### D.2  TEMPORAL TAMPER

We consider three temporal tamper methods: Frame Drop, Frame Insert, and Frame Swap. The first two types of tamper result in changes to both the total number of frames and their order, while the latter only alters the frame order.

- **Frame Drop** involves randomly selecting a position and removing the entire frame at that location.
- **Frame Insert** involves choosing a random location and inserting an additional frame, either by selecting the adjacent frame or generating a new one with Gaussian noise.
- **Frame Swap** involves selecting two frames at random and swapping their positions.

In our experiments, for the first two tamper methods, we randomly choose a location and execute the corresponding operation: $Op(\text{frame}_p)$, where $p \sim \{1, \ldots, f\}$ and $Op \in \{\text{Drop}, \text{Insert}\}$. For Frame Insert, the inserted frame is randomly selected from the adjacent frame or generated with Gaussian noise. For Frame Swap, we start from a designated position and swap the frame at that position with the adjacent frame at regular intervals: $\text{Swap}(\text{frame}_p, \text{frame}_{p+1})$, where $p = 2 + 4k$ and $k \sim \{0, 1, 2, \ldots, \lfloor \frac{f-3}{4} \rfloor\}$.

### D.3 SPATIAL TAMPER

We consider two types of spatial tamper on generated videos: Crop&Drop and Inpainting. Crop&Drop involves randomly cropping a portion of the video content (ratio = 0.5) and flipping the remaining part, or randomly dropping a portion of the content (ratio = 0.5) and flipping that part. Crop&Drop simulates outpainting (background changes) and inpainting (subject changes), respectively, making it useful for large-scale tamper localization tests on all generated videos. To make the tests more reflective of real-world application scenarios, we also manually mask the main objects in some videos and use inpainting models for tamper. Specifically, we employ STTN (Zeng et al., 2020) and ProPainter (Zhou et al., 2023), two video inpainting models capable of removing objects from the masked locations, thereby achieving video tamper.

### D.4 GENERALITY TEST

**ZeroScope.** We generate 200 videos at a resolution of 256 using the prompts from ModelScope for watermark evaluation, following the same approach. Default sampling settings are applied, with 50 inference and inversion steps, configuring $k_f, k_c, k_h, k_w$ to 8, 1, 4, 4 to embed 512 bits. For tamper localization, we select 40 videos and consider all temporal tamper types along with the Crop&Drop spatial tamper on generated videos, as described in Sec. D.3. For the thresholds $t_{wm}, t_{orig}$, and the hierarchical level $L$, we use the same settings as ModelScope.

**I2VGen-XL.** We generate 200 videos at a resolution of 512 using images generated by Stable Diffusion 2.1 for watermark evaluation, following the same approach with Stable-Video-Diffusion. Default sampling settings are applied, with 50 inference and inversion steps, configuring $k_f, k_c, k_h, k_w$ to 8, 1, 8, 8 to embed 512 bits. For tamper localization, we select 40 videos and consider all temporal tamper types along with the Crop&Drop spatial tamper on generated videos, as described in Sec. D.3. For the thresholds $t_{wm}, t_{orig}$, and the hierarchical level $L$, we use the same settings as Stable-Video-Diffusion.

**Three Versions of Stable Diffusion.** We randomly sample 500 prompts from the Stable-Diffusion-Prompts dataset[2] to generate 500 images at a resolution of 512. Default sampling configurations are applied with 50 inference and inversion steps. For spatial tamper localization, we consider the Crop&Drop method. Specifically, for Crop, the areas outside the cropped region are turned black, and for Drop, the removed parts are replaced with black. For the thresholds $t_{wm}, t_{orig}$, and the hierarchical level $L$, we use the same values as in ModelScope.

### D.5 DISTORTIONS FOR TESTING ROBUSTNESS

**Watermark extraction.** We consider the following distortions: three video distortions—MPEG-4, Frame Average ($N = 3$), and Frame Swap ($p = 0.25$)—and eight image distortions applied to each video frame: Gaussian Noise ($\sigma = 0.1$), Gaussian Blur ($r = 4$), Median Blur ($k = 7$), Salt and Pepper Noise ($p = 0.1$), Random Crop (0.5), Random Drop (0.5), Resize (0.3), and Brightness (factor = 6).

**Spatial tamper localization.** We consider the following distortions: MPEG-4, Frame Average ($N = 3$), Gaussian Noise ($\sigma = 0.1$), Gaussian Blur ($r = 4$), Median Blur ($k = 7$), Salt and Pepper Noise ($p = 0.1$), Random Crop (0.5), Random Drop (0.5), Resize (0.3), and Brightness (factor = 6).

---

[2]https://huggingface.co/datasets/Gustavosta/Stable-Diffusion-Prompts

# E    MORE EXPERIMENTAL RESULTS

## E.1    VIDEO QUALITY

**Objective metrics.**    We provide the specific metric values of video quality in Table 9. As shown in the table, the watermarking method with post-processing causes more significant degradation to the videos in terms of Dynamic Degree and Imaging Quality. However, for Background Consistency and Motion Smoothness, videos with post-processing sometimes show a slight increase in range. We believe this is due to the watermark embedding introducing some blurring to the video, resulting in better smoothness, which leads to a slight "anomalous" increase in continuity metrics.

Table 9: Specific metric values for video quality assessment.

| Method | Subject Consistency | Background Consistency | Motion Smoothness | Dynamic Degree | Imaging Quality |
|---|---|---|---|---|---|
| ModelScope | | | | | |
| RivaGAN | 0.922 | 0.951 | 0.960 | 0.540 | 0.648 |
| MBRS | 0.923 | 0.951 | 0.961 | 0.540 | 0.643 |
| CIN | 0.922 | 0.964 | 0.971 | 0.520 | 0.402 |
| PIMoG | 0.920 | 0.960 | 0.972 | 0.510 | 0.405 |
| SepMark | 0.919 | 0.952 | 0.961 | 0.520 | 0.646 |
| VideoShield | 0.923 | 0.954 | 0.961 | 0.545 | 0.648 |
| Stable-Video-Diffusion | | | | | |
| RivaGAN | 0.938 | 0.968 | 0.957 | 0.710 | 0.607 |
| MBRS | 0.938 | 0.968 | 0.960 | 0.693 | 0.584 |
| CIN | 0.937 | 0.973 | 0.965 | 0.672 | 0.431 |
| PIMoG | 0.936 | 0.971 | 0.965 | 0.672 | 0.427 |
| SepMark | 0.936 | 0.968 | 0.959 | 0.650 | 0.585 |
| VideoShield | 0.939 | 0.968 | 0.957 | 0.710 | 0.607 |

**User study.**    We further conduct a user study to perform a subjective comparison of video quality. We randomly select 10 videos from each of the 200 videos generated by MS and SVD. After watermarking these videos by different methods, we distribute the watermarked videos to 24 distinct users with the instruction: "Please choose the video that you consider to have the highest quality, based on Subject Consistency, Background Consistency, Motion Smoothness, Dynamic Degree, and Imaging Quality."

The results are shown in Table 10. As observed, VIDEOSHIELD receives the most votes for both MS and SVD, followed closely by RivaGAN with a slight gap, while MBRS also garners relatively high votes. Conversely, CIN, PIMoG, and SepMark receive very few votes, indicating more significant visible degradation in video quality.

Table 10: Subjective evaluation results for video quality comparison.

| Method | RivaGAN | MBRS | CIN | PIMoG | SepMark | VIDEOSHIELD |
|---|---|---|---|---|---|---|
| MS | 78 | 59 | 3 | 2 | 6 | 92 |
| SVD | 82 | 51 | 1 | 0 | 8 | 98 |

## E.2    WATERMARK ROBUSTNESS

The detailed watermark extraction accuracy of various watermarking methods under different distortions is provided in Table 11, offering a comprehensive comparison of their performance across multiple scenarios.

Table 11: Specific extraction accuracy of various watermarking methods under different distortions.

| Method | MPEG-4 | Frame Average | Frame Swap | Gaussian Noise | Gaussian Blur | Median Blur | Salt and Pepper Noise | Random Crop | Random Drop | Resize | Brightness | Average |
|---|---|---|---|---|---|---|---|---|---|---|---|---|
| ModelScope | | | | | | | | | | | | |
| RivaGAN | 0.963 | 0.974 | 0.993 | 0.870 | 0.825 | 0.978 | 0.809 | 0.977 | 0.992 | 0.984 | 0.791 | 0.923 |
| MBRS | 0.996 | 1.000 | 1.000 | 0.714 | 0.501 | 0.500 | 0.669 | 0.776 | 0.974 | 0.603 | 0.822 | 0.778 |
| CIN | 0.785 | 1.000 | 1.000 | 0.899 | 0.499 | 0.783 | 0.842 | 1.000 | 1.000 | 0.900 | 0.973 | 0.880 |
| PIMoG | 0.903 | 1.000 | 1.000 | 0.577 | 0.817 | 0.987 | 0.566 | 0.959 | 0.990 | 0.991 | 0.957 | 0.886 |
| SepMark | 0.999 | 0.999 | 0.998 | 0.977 | 0.709 | 0.805 | 0.978 | 0.985 | 0.999 | 0.954 | 0.840 | 0.931 |
| **VIDEOSHIELD** | 0.999 | 0.996 | 1.000 | 0.946 | 0.994 | 0.999 | 0.907 | 0.996 | 1.000 | 0.999 | 0.990 | 0.984 |
| Stable-Video-Diffusion | | | | | | | | | | | | |
| RivaGAN | 0.949 | 0.922 | 0.989 | 0.667 | 0.791 | 0.967 | 0.747 | 0.968 | 0.987 | 0.973 | 0.829 | 0.890 |
| MBRS | 0.961 | 0.999 | 0.999 | 0.580 | 0.498 | 0.584 | 0.637 | 0.778 | 0.974 | 0.633 | 0.897 | 0.776 |
| CIN | 0.798 | 1.000 | 1.000 | 0.663 | 0.504 | 0.782 | 0.881 | 1.000 | 1.000 | 0.872 | 0.997 | 0.863 |
| PIMoG | 0.949 | 0.999 | 0.999 | 0.541 | 0.755 | 0.979 | 0.593 | 0.952 | 0.986 | 0.981 | 0.909 | 0.877 |
| SepMark | 0.986 | 0.998 | 0.998 | 0.809 | 0.727 | 0.859 | 0.928 | 0.986 | 0.998 | 0.936 | 0.934 | 0.924 |
| **VIDEOSHIELD** | 0.984 | 0.986 | 0.999 | 0.768 | 0.972 | 0.995 | 0.897 | 0.974 | 0.998 | 0.995 | 0.984 | 0.959 |

### E.3 Impact of different inference and inversion configurations

We provide the detailed results under different inference and inversion configurations in Table 12.

Table 12: Results of different configurations in terms of three test dimensions. Since Stable-Video-Diffusion cannot generate high-quality video using a non-default scheduler, we omit this part of the test. Gray cells denote the default setting.

| Dimension | ModelScope | | | | | | | | | | | | | |
| | Inference Step | | | Inversion Step | | | Text Guidance | | | Scheduler | | | | |
| | 10 | 25 | 50 | 10 | 25 | 50 | 6 | 9 | 12 | DDIM | UniPC | PNDM | DEIS | DPM |
|---|---|---|---|---|---|---|---|---|---|---|---|---|---|---|
| Watermark | 1.000 | 1.000 | 1.000 | 1.000 | 1.000 | 1.000 | 1.000 | 1.000 | 1.000 | 1.000 | 1.000 | 1.000 | 1.000 | 0.998 |
| Temporal | 1.000 | 1.000 | 1.000 | 1.000 | 1.000 | 1.000 | 1.000 | 1.000 | 1.000 | 1.000 | 1.000 | 1.000 | 1.000 | 0.981 |
| Spatial | 0.906 | 0.906 | 0.904 | 0.895 | 0.906 | 0.909 | 0.907 | 0.906 | 0.902 | 0.906 | 0.902 | 0.907 | 0.902 | 0.786 |

| Dimension | Stable-Video-Diffusion | | | | | | | | | | | | | |
| | Inference Step | | | Inversion Step | | | Image Guidance | | | Scheduler | | | | |
| | 10 | 25 | 50 | 10 | 25 | 50 | 2 | 3 | 4 | Euler | - | - | - | - |
|---|---|---|---|---|---|---|---|---|---|---|---|---|---|---|
| Watermark | 0.997 | 0.999 | 0.999 | 0.999 | 0.999 | 0.999 | 0.997 | 0.999 | 0.985 | 0.999 | - | - | - | - |
| Temporal | 0.929 | 0.936 | 0.969 | 0.934 | 0.936 | 0.937 | 0.972 | 0.936 | 0.866 | 0.936 | - | - | - | - |
| Spatial | 0.730 | 0.761 | 0.783 | 0.761 | 0.761 | 0.763 | 0.810 | 0.761 | 0.718 | 0.761 | - | - | - | - |

### E.4 Watermark Extraction from Spatial Tampered Videos

We evaluate the accuracy of watermark extraction after the video undergoes spatial tampering. As shown in Table 13, even after various types of spatial tampering, the watermark extraction accuracy remains close to 100%, demonstrating its effectiveness in supporting forensics after video tampering.

Table 13: Watermark extraction accuracy on spatial tampered videos.

| Model | STTN | ProPainter | Crop&Drop |
|---|---|---|---|
| MS | 0.975 | 0.971 | 0.997 |
| SVD | 0.976 | 0.975 | 0.982 |

### E.5 Robustness against more distortions

To further evaluate the resilience of VIDEOSHIELD, we test its robustness against a range of video compression methods (including extreme compression) and color adjustments, as shown in Table 14. *For video compression*, the typical CRF (Constant Rate Factor) range is 18–28, where 18 corresponds to higher video quality and 28 to lower quality. The results show that only under extreme compression conditions (CRF > 28) do the watermark extraction and localization performance for both MS and SVD start to degrade significantly. However, adjusting the threshold substantially improves spatial localization performance. At this point, the video quality is already severely degraded, making the performance drop reasonable. *For color adjustments*, VIDEOSHIELD shows strong robustness, maintaining high performance in both watermark extraction and tamper localization. Overall, VIDEOSHIELD demonstrates good robustness, making it suitable for most real-world scenarios.

### E.6 Comparison with EDITGUARD

We conduct comprehensive experiments to compare VIDEOSHIELD with EditGuard (Zhang et al., 2023b), a recently open-sourced proactive tamper localization model known for its outstanding performance. Our evaluation encompasses both videos and images generated by various models. Specifically, we employ the default spatial tamper method, Crop&Drop, to create 200 tampered videos for each video generation model and 500 tampered images for each image generation model. As shown in Table 15, VIDEOSHIELD outperforms EditGuard on five models, with the exception of SVD and I2VGen. The relatively lower performance on SVD, in particular, is discussed in Appendix G.3, where we attribute this to the quality of the generated content.

Table 14: Performance under more spatial distortions. As discussed in Sec. 3.5, 'w/o' refers to using $t_{wm}$ and $t_{orig}$ derived from the clean watermarked videos, while 'w/' indicates that various distortions are introduced during the distribution testing, leading to adjustments in $t_{wm}$ and $t_{orig}$. Hue=0.5 is the max factor we can configure.

| Dimension | Model | Clean | H.264 (CRF) | | | | | Hue=0.5 |
| | | | 18 | 23 | 28 | 33 | 38 | |
| --- | --- | --- | --- | --- | --- | --- | --- | --- |
| Watermark | MS | 1.000 | 0.999 | 0.999 | 0.997 | 0.978 | 0.907 | 1.000 |
| | SVD | 0.999 | 0.988 | 0.980 | 0.961 | 0.933 | 0.888 | 0.993 |
| Spatial | MS w/o | 0.906 | 0.861 | 0.826 | 0.746 | 0.639 | 0.555 | 0.842 |
| | MS w/ | 0.806 | 0.814 | 0.812 | 0.788 | 0.720 | 0.627 | 0.815 |
| | SVD w/o | 0.761 | 0.667 | 0.635 | 0.607 | 0.577 | 0.547 | 0.684 |
| | SVD w/ | 0.708 | 0.718 | 0.699 | 0.664 | 0.639 | 0.605 | 0.724 |

Table 15: Comparison of spatial tamper localization performance between VIDEOSHIELD and EditGuard on videos and images generated by various models. SD stands for Stable Diffusion. The table presents the average values of five metrics: F1, Precision, Recall, AUC, and IoU.

| Method | Video | | | | Image | | |
| | MS | SVD | ZS | I2VGen | SD 1.4 | SD 1.5 | SD 2.1 |
| --- | --- | --- | --- | --- | --- | --- | --- |
| EditGuard | 0.890 | 0.886 | 0.880 | 0.882 | 0.885 | 0.885 | 0.885 |
| VIDEOSHIELD | 0.906 | 0.761 | 0.900 | 0.867 | 0.938 | 0.939 | 0.928 |

### E.7 COMPUTATION OVERHEAD

**Results.** We provide the computation overhead of VIDEOSHIELD in Table 16. The primary GPU memory usage and runtime overhead are concentrated in the DDIM inversion stage. However, as shown in the table for step = 10 and step = 25, reducing the number of inversion steps can significantly decrease the runtime, with only a slight sacrifice in performance as shown in Table 12.

Table 16: Computational overhead of VIDEOSHIELD in terms of GPU memory usage and runtime, evaluated on a single NVIDIA RTX A6000 GPU (49 GB) in FP16 mode. The average runtime of different stages is reported based on 50 generated videos, with the batch size set to 1. "Step" refers to the inversion step, while "Watermark," "Spatial," and "Temporal" indicate the average runtime of watermark extraction, spatial tamper localization, and temporal tamper localization, respectively.

| Model | #Params | Resolution | GPU (GB) | Runtime (s) | | | | |
| | | | | Step=10 | Step=25 | Watermark | Spatial | Temporal |
| --- | --- | --- | --- | --- | --- | --- | --- | --- |
| MS | 1.83B | 256 | 3.77 | 1.2408 | 3.0617 | 0.0011 | 0.0019 | 0.0004 |
| SVD | 2.25B | 512 | 5.32 | 4.3214 | 10.2027 | 0.0023 | 0.0019 | 0.0004 |
| ZS | 1.83B | 256 | 3.77 | 1.2430 | 3.0492 | 0.0011 | 0.0019 | 0.0004 |
| I2VGen | 2.48B | 512 | 5.99 | 4.6700 | 11.1526 | 0.0023 | 0.0019 | 0.0004 |

**Potential speed-up methods.** In practice, the number of inversion steps required for diffusion model inversion aligns best with the number of inference steps. Existing distillation techniques can effectively reduce the inference steps of diffusion models, thereby substantially lowering the inference cost of VIDEOSHIELD. To explore potential efficiency improvements, we conduct experiments on SD 1.5 and SD 1.5 Turbo (the distilled version of SD 1.5) for image tamper localization, with the results summarized in Table 17. These experiments show that distillation techniques significantly reduce the number of required inversion steps while maintaining comparable localization performance. This suggests that similar distillation techniques can be adapted for video diffusion models, offering a promising approach to significantly reduce the computational overhead of VIDEOSHIELD.

## F MORE ANALYSIS

**Different distributions based on $k$ in $PTB$.** Table 18 shows that when $k$ is set to 100, there are a few outliers, leading to a wider statistical distribution range. For example, in ModelScope

Table 17: Comparison of inversion runtime reduction in the SD 1.5 Turbo model without compromising the performance of VIDEOSHIELD. SD 1.5 Turbo is a distilled version of SD 1.5, which enables image generation in significantly fewer steps, potentially even a single step.

| Model | Step | Runtime (s) | Spatial Localization |
|---|---|---|---|
| SD1.5 | 10 | 0.3146 | 0.912 |
| SD1.5 Turbo | 2 | 0.0874 | 0.909 |

with $\mu = 2$, when $k = 100$, $\mathcal{A}_{wm}$ and $\mathcal{A}_{orig}$ range from [0.28, 1.00] and [0.06, 0.84], respectively; whereas for $k = 99$, $\mathcal{A}_{wm}$ and $\mathcal{A}_{orig}$ are [0.56, 0.93] and [0.28, 0.68], respectively. Clearly, $k = 100$ significantly increases the overlap compared to $k = 99$, and the thresholds ($t_{wm}$ and $t_{orig}$) obtained may cause many points to become indistinguishable, thereby reducing localization performance.

Table 18: The specific distribution of the watermarked and the original videos corresponding to different values of $k$ in $PTB$. We **bold** the distribution range used for calculating the default thresholds $t_{wm}$ and $t_{orig}$.

| Distribution | Local Value | ModelScope | | | | Stable-Video-Diffusion | | | |
|---|---|---|---|---|---|---|---|---|---|
| | | 97 | 98 | 99 | 100 | 97 | 98 | 99 | 100 |
| $\mathcal{A}_{wm}$ | 1 | [0.25, 1.00] | [0.25, 1.00] | **[0.25, 1.00]** | [0.00, 1.00] | [0.25, 1.00] | **[0.00, 1.00]** | [0.00, 1.00] | [0.00, 1.00] |
| | 2 | [0.62, 0.90] | [0.59, 0.93] | **[0.56, 0.93]** | [0.28, 1.00] | [0.40, 0.81] | **[0.37, 0.81]** | [0.34, 0.84] | [0.12, 1.00] |
| | 4 | [0.69, 0.85] | [0.68, 0.86] | **[0.66, 0.88]** | [0.55, 0.96] | [0.48, 0.73] | **[0.47, 0.74]** | [0.46. 0.76] | [0.38, 0.87] |
| $\mathcal{A}_{orig}$ | 1 | [0.00, 1.00] | [0.00, 1.00] | **[0.00, 1.00]** | [0.00, 1.00] | [0.00, 1.00] | **[0.00, 1.00]** | [0.00, 1.00] | [0.00, 1.00] |
| | 2 | [0.34, 0.65] | [0.31, 0.68] | **[0.28, 0.68]** | [0.06, 0.84] | [0.34, 0.65] | **[0.31, 0.68]** | [0.28, 0.71] | [0.12, 0.87] |
| | 4 | [0.44, 0.55] | [0.43, 0.56] | **[0.42, 0.57]** | [0.37, 0.62] | [0.44, 0.55] | **[0.43, 0.56]** | [0.42, 0.57] | [0.36, 0.63] |

**Accuracy distributions of different local values for seven Models.** As shown in Figure 9, we observe two key points: (1) The higher the comparison accuracy distribution of a model, the better its spatial tamper localization performance. (2) The distributions for models of the same type (T2V, I2V, and T2I) are similar. These conclusions are consistent with the results presented in Table 6 and Table 7.

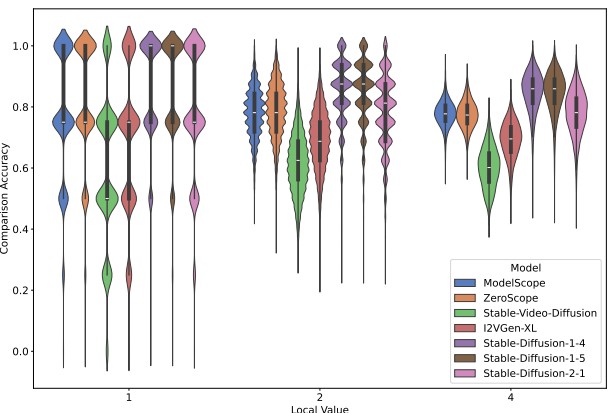

Figure 9: Comparison accuracy distributions of different local values on the watermarked and original videos generated by seven models.

**The relationship between VIDEOSHIELD's performance and model's inversion accuracy.** We further test the average inversion accuracy of the four video generation models we test. As mentioned before, the performance of VIDEOSHIELD on different models varies. Since the method is based on DDIM Inversion, we speculate that this difference is related to the inversion accuracy of the model itself and conduct corresponding tests. Figure 10 shows the relationship between VIDEOSHIELD's performance and average inversion accuracy. It can be seen that the performance of our method is positively correlated with inversion accuracy. The higher inversion accuracy is, the better the performance of our method is.

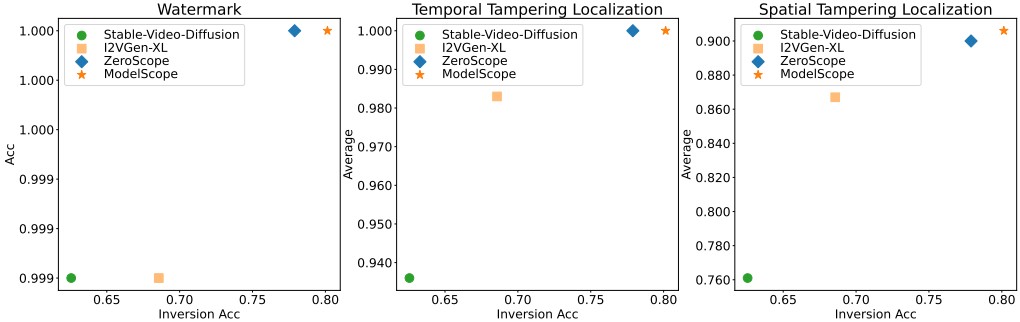

Figure 10: The relationship between inversion accuracy and VIDEOSHIELD's performance in three dimensions. The inversion acc refers to the average accuracy of all points in the comparison bits matrix $Cmp$.

**The relationship between VIDEOSHIELD's performance and different resolutions.** We further investigate the performance of VIDEOSHIELD on videos with different resolutions. Since ZeroScope is trained on videos of various resolutions, we choose it to generate videos at different resolutions for this analysis. As shown in Table 19, VIDEOSHIELD 's spatial tamper localization performance improves as the resolution increases. We believe this is because higher video resolutions capture more details and provide better quality, allowing the template bits to better integrate into the generated video. This, in turn, enhances the inversion accuracy, leading to improved spatial tamper localization performance.

Table 19: Performance on videos generated by ZeroScope with different resolutions.

| Resolution | 256 | 320 | 384 | 448 |
|---|---|---|---|---|
| Watermark | 1.000 | 1.000 | 1.000 | 1.000 |
| Temporal | 1.000 | 1.000 | 1.000 | 1.000 |
| Spatial | 0.900 | 0.918 | 0.941 | 0.948 |

**Importance of adjusting $t_{wm}$&$t_{orig}$ of $PTB$ and $L$ of *HSTR*, when performing spatial tamper localization on distorted tampered videos.** As shown in Figure 11, when distortion is added to the watermarked video, its distribution $\mathcal{A}_{wm}$ shifts towards $\mathcal{A}_{orig}$, reducing the distinction between tampered and original regions. To mitigate the impact of this shift, we can readjust $t_{wm}$ and $t_{orig}$ derived from $\mathcal{A}_{wm}$ and $\mathcal{A}_{orig}$. As depicted in Figure 13, adjusting the threshold effectively enhances the robustness of spatial tamper localization against various distortions, although it slightly reduces performance on clean tampered videos. Additionally, as shown in Figure 12, we re-test the spatial tamper localization performance corresponding to different $L$ values under distortion scenarios using adjusted thresholds. We find that for Stable-Video-Diffusion, using a larger $L = 4$ for tampered videos with added distortions yields better results. We believe this is because the inversion accuracy of Stable-Video-Diffusion is inherently low; thus, after adding distortions, more comparison bits are needed for a more accurate assessment, which enhances robustness.

## G  MORE VISUAL EXAMPLES

### G.1  MAIN RESULTS

**Spatial tamper localization results under different hierarchical level $L$.** See Figure 14.

**More spatial tamper localization results on videos generated by ModelScope and Stable-Video-Diffusion for different types of spatial tamper.** See Figure 15, Figure 16, Figure 17 and Figure 18.

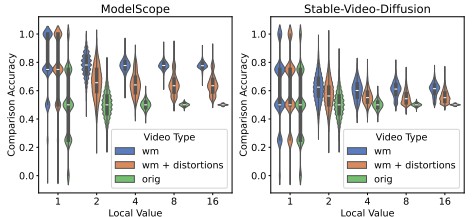 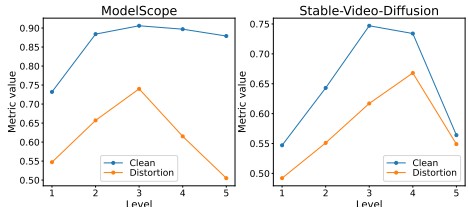

Figure 11: Comparison accuracy distributions $\mathcal{A}$ of different local values $\mu$ on the original, watermarked and distorted watermarked videos.

Figure 12: The line chart depicting the variation of spatial tamper localization performance as hierarchical level $L$ changes on clean and distorted watermarked videos.

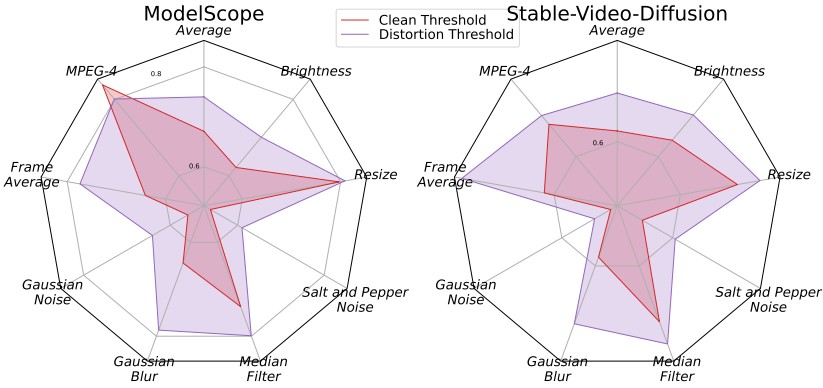

Figure 13: Spatial localization performance against distortions using clean and distortion threshold.

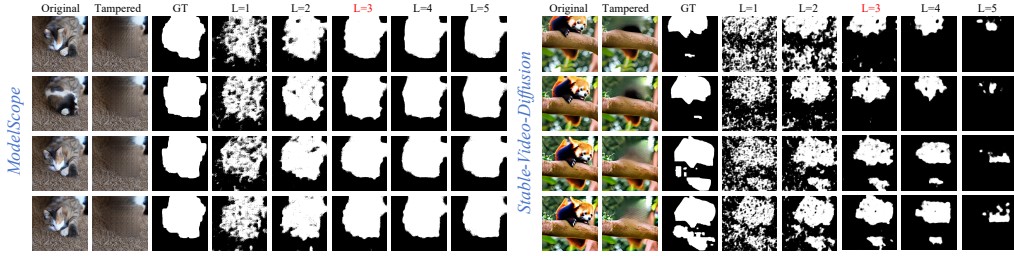

Figure 14: Some visual examples of spatial tamper localization with different hierarchical level $L$.

## G.2 GENERALITY

**Spatial tamper localization results on videos generated by other video models.** See Figure 19 and Figure 20.

**Spatial tamper localization results on images generated by different T2I models.** See Figure 21.

## G.3 FAILURE CASES OF STABLE-VIDEO-DIFFUSION

**Limitations of the video model restrict localization performance.** As shown in Figure 22 and Figure 23, the residual image indicates that the content in the areas where spatial tamper localization fails (highlighted by the white regions of the predicted mask, which are not actually tampered) closely resembles that of the conditional image (represented by the black areas in the residual). Consequently, the noise corresponding to the template bits in these regions has not been effectively denoised, resulting in content that is merely a replica of the conditional image. The inverted bits ob-

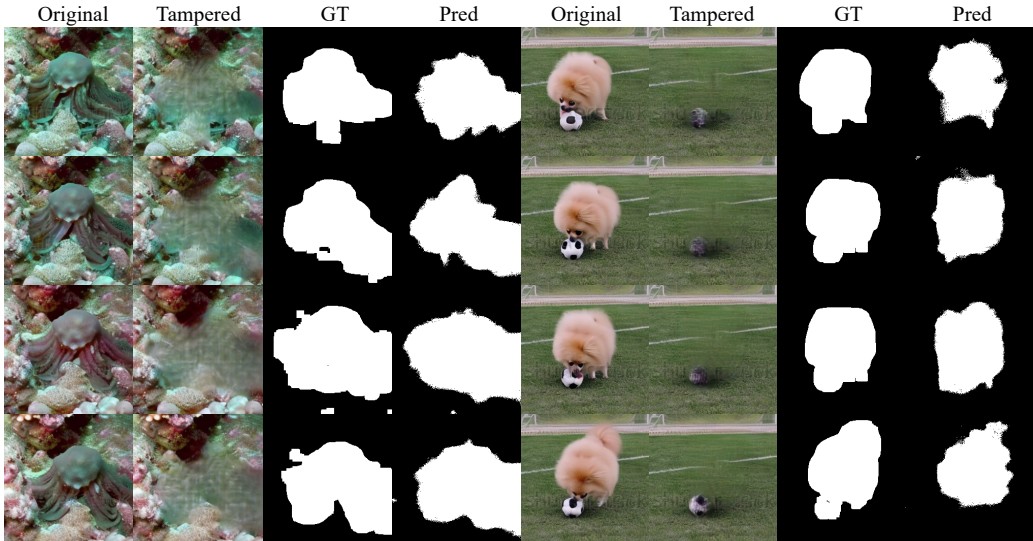

Figure 15: Some visual spatial tamper localization results on videos generated by ModelScope with STTN tamper.

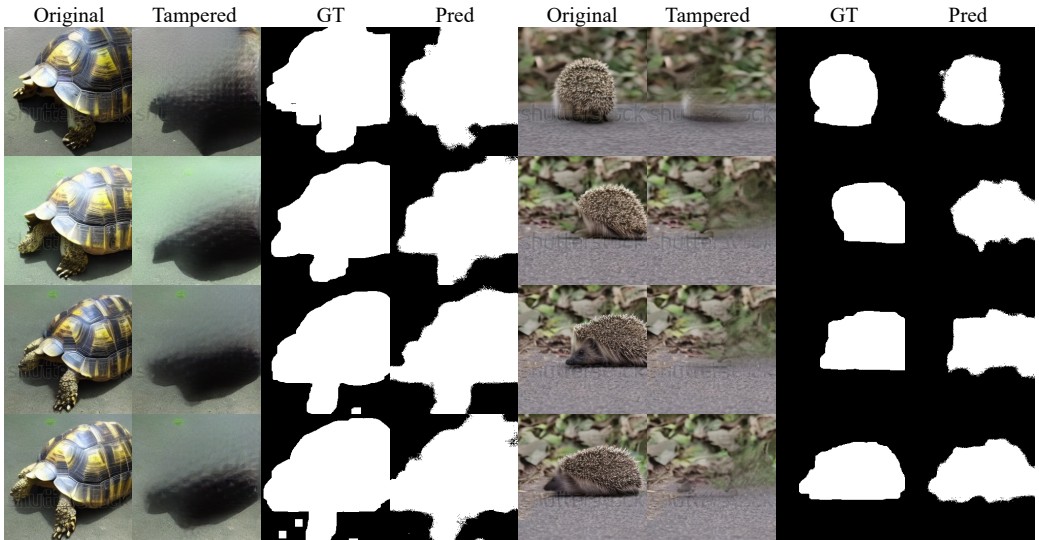

Figure 16: Some visual spatial tamper localization results on videos generated by ModelScope with ProPainter tamper.

tained through inversion clearly do not match the original template bits, leading to the entire region being classified as tampered and ultimately reducing localization accuracy. Thus, this limitation stems from the model itself, which struggles to generate a video where the overall content can move seamlessly.

**Enhancing localization performance by improving video quality through sampling configuration adjustments.** As shown in Figure 24, adjusting the sampling configuration allows the model to generate more original content in the video instead of merely copying from the conditional images. This not only enhances video quality but also enables better integration of the noise corresponding to the template bits with the relevant video content. Consequently, this leads to improved reconstruction through inversion, resulting in better localization performance when compared to the original template bits.

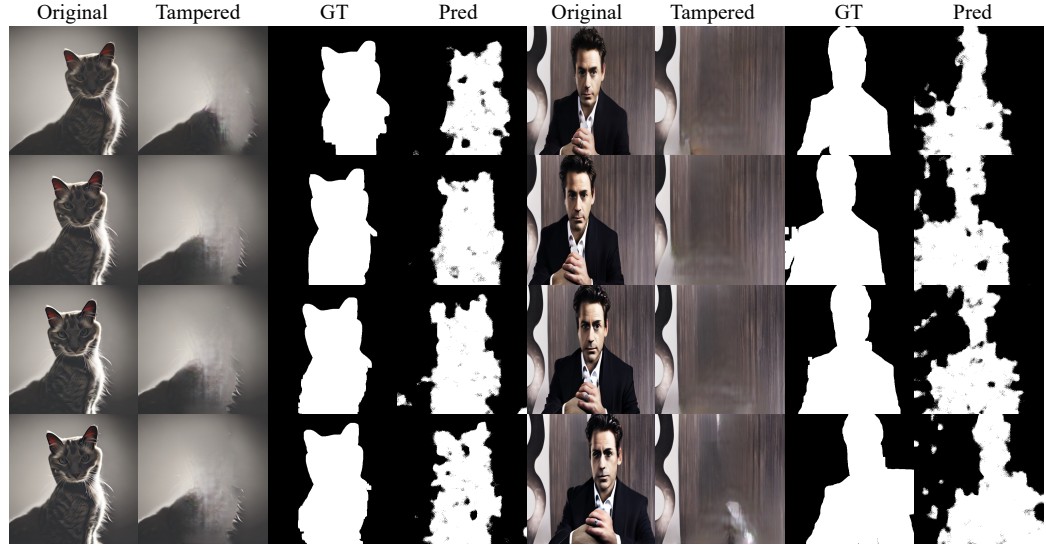

Figure 17: Some visual spatial tamper localization results on videos generated by Stable-Video-Diffusion with STTN tamper.

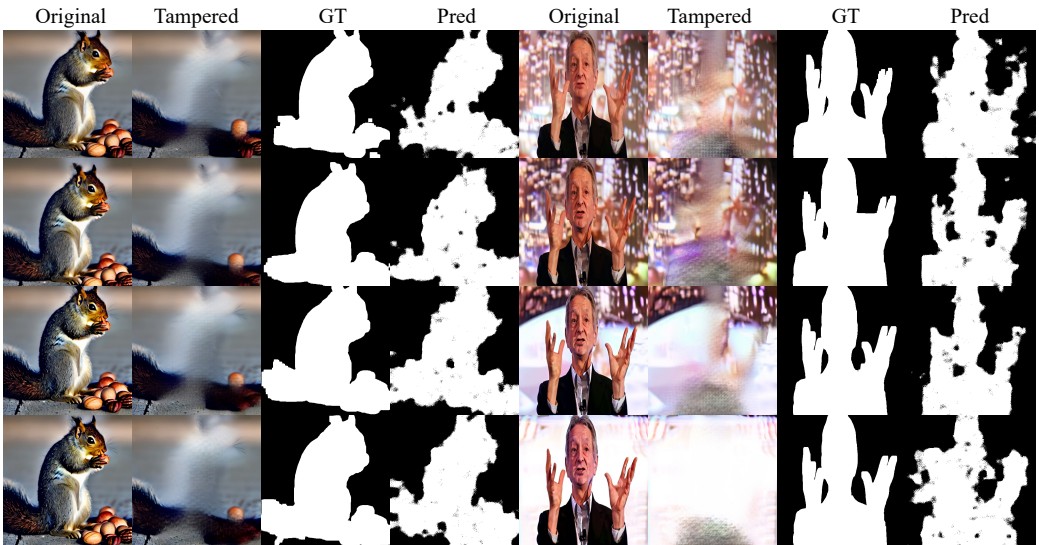

Figure 18: Some visual spatial tamper localization results on videos generated by Stable-Video-Diffusion with ProPainter tamper.

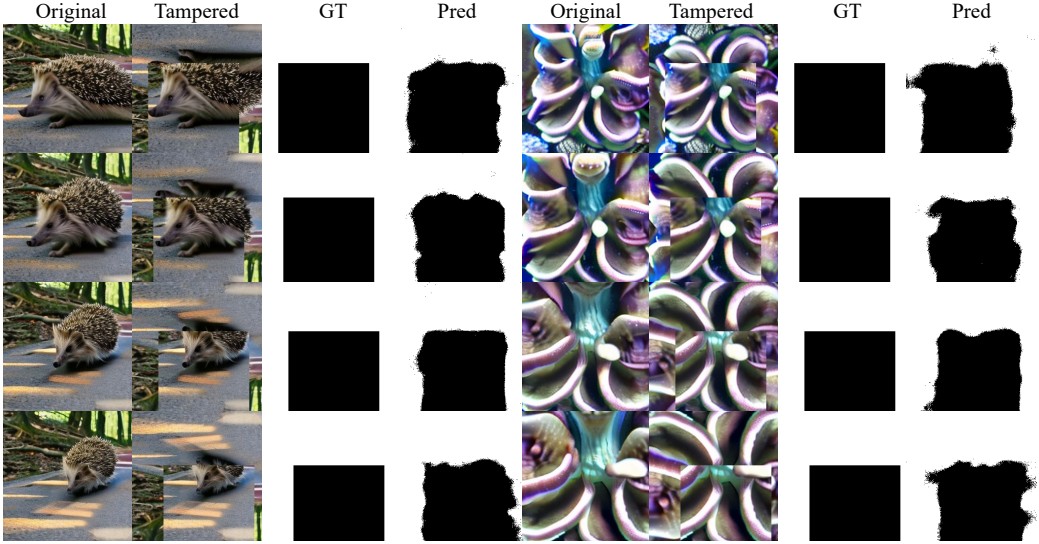

Figure 19: Some visual spatial tamper localization results on videos generated by ZeroScope with Crop&Drop tamper.

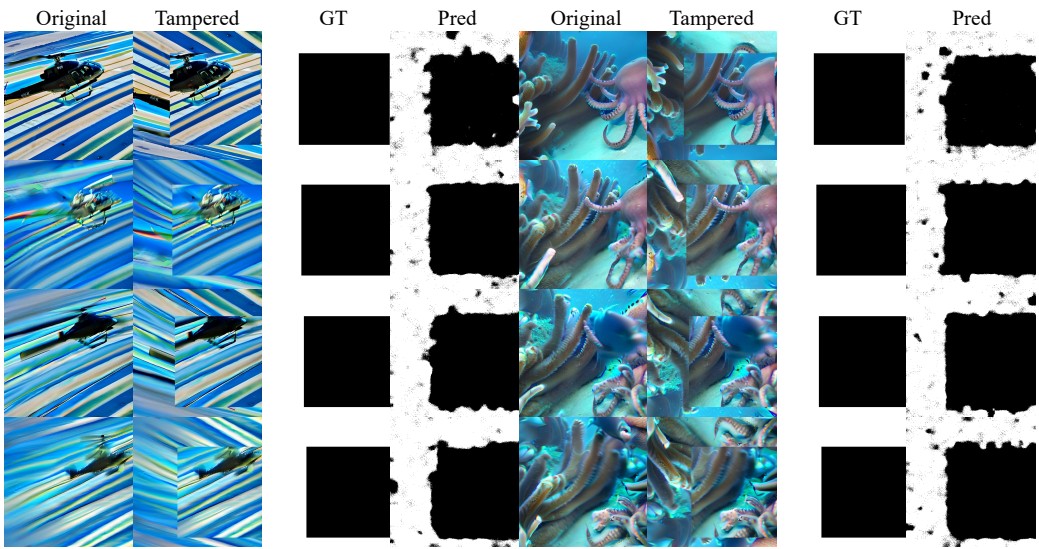

Figure 20: Some visual spatial tamper localization results on videos generated by I2VGen-XL with Crop&Drop tamper.

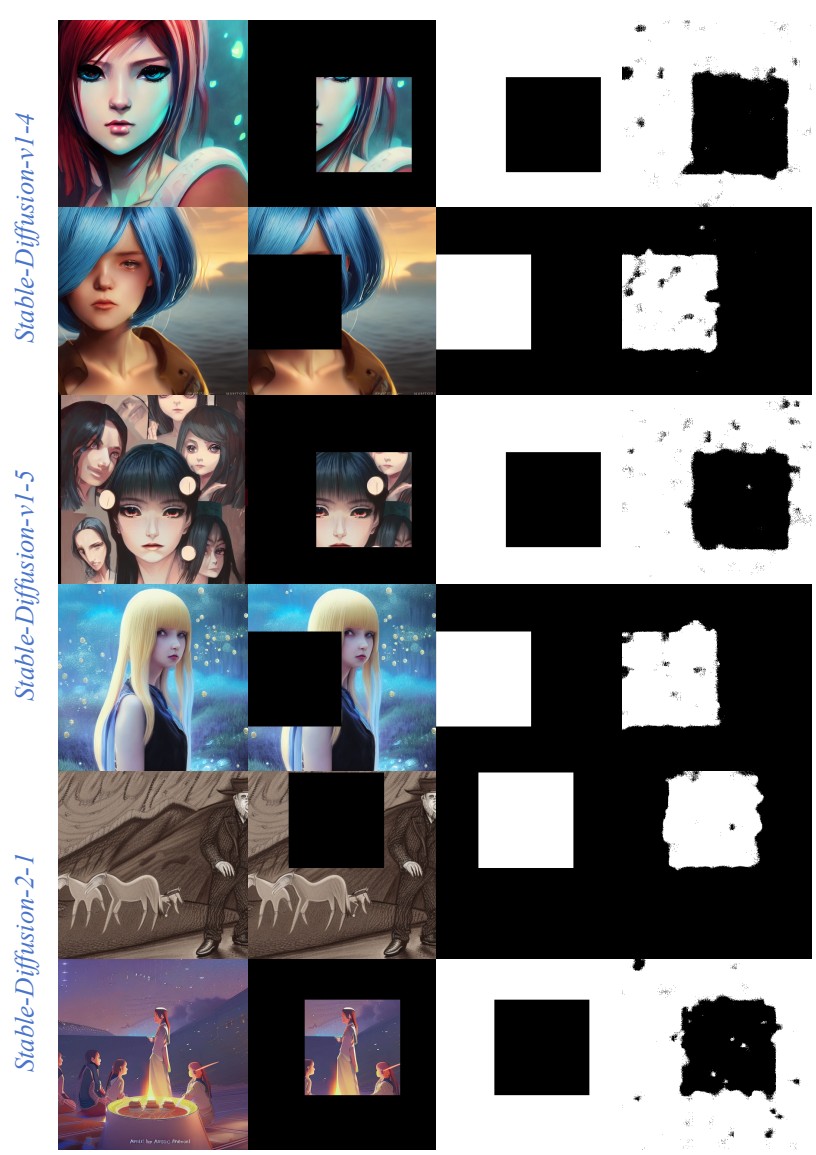

Figure 21: Some spatial tamper localization results on images generated by different versions of Stable Diffusion with Crop&Drop tamper.

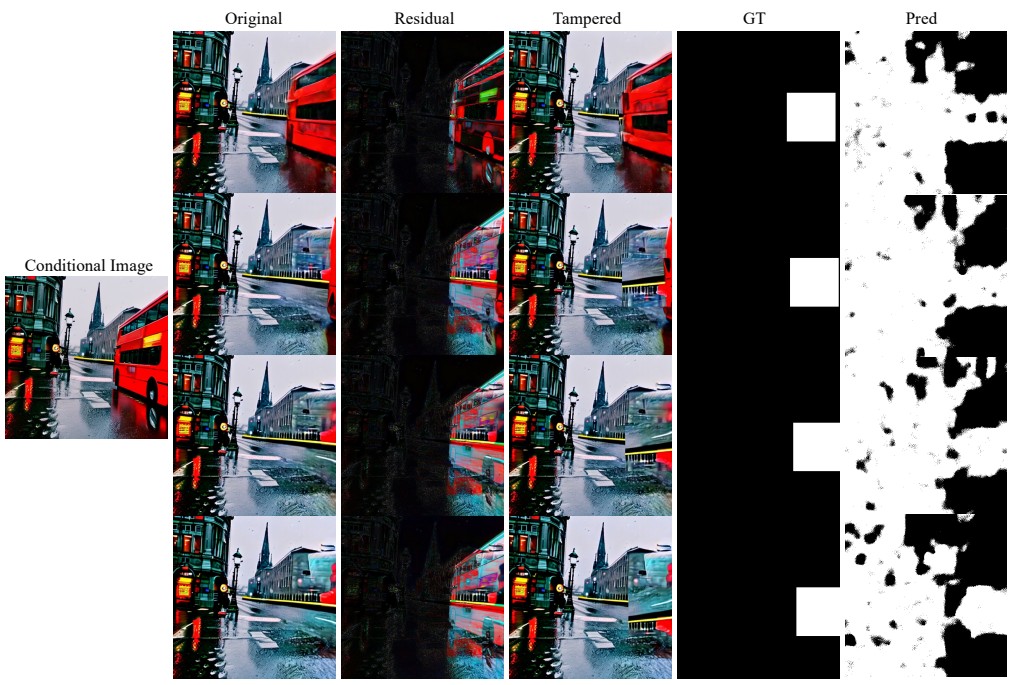

Figure 22: Examples of poor spatial tamper localization on videos generated using Stable-Video-Diffusion. "Residual" refers to the difference obtained by subtracting the frames of the video from the conditional image.

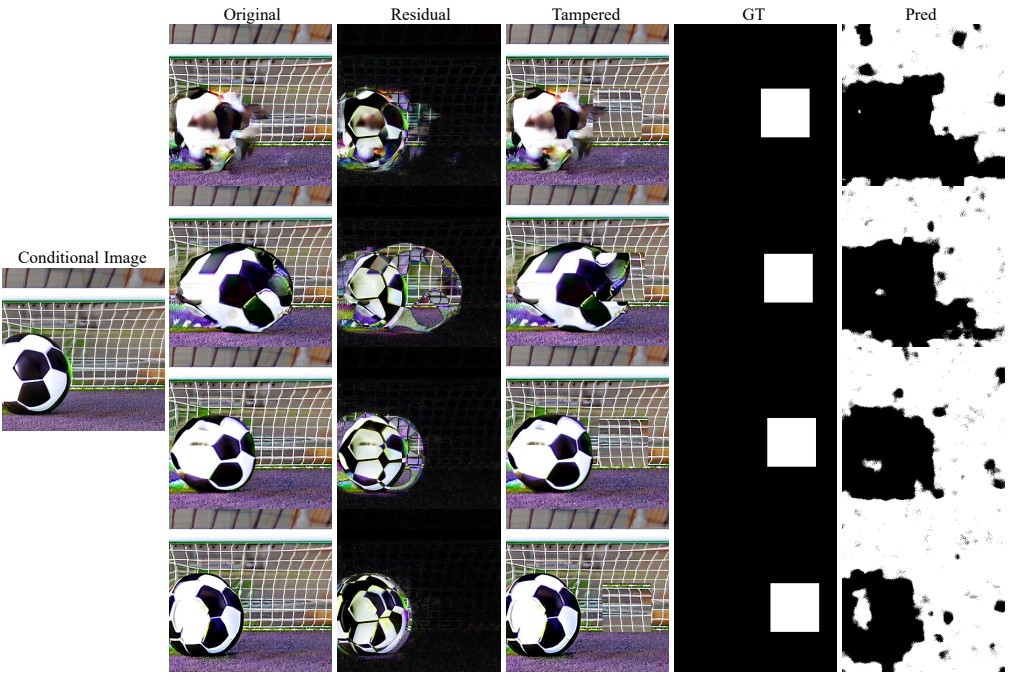

Figure 23: Examples of poor spatial tamper localization on videos generated using Stable-Video-Diffusion.

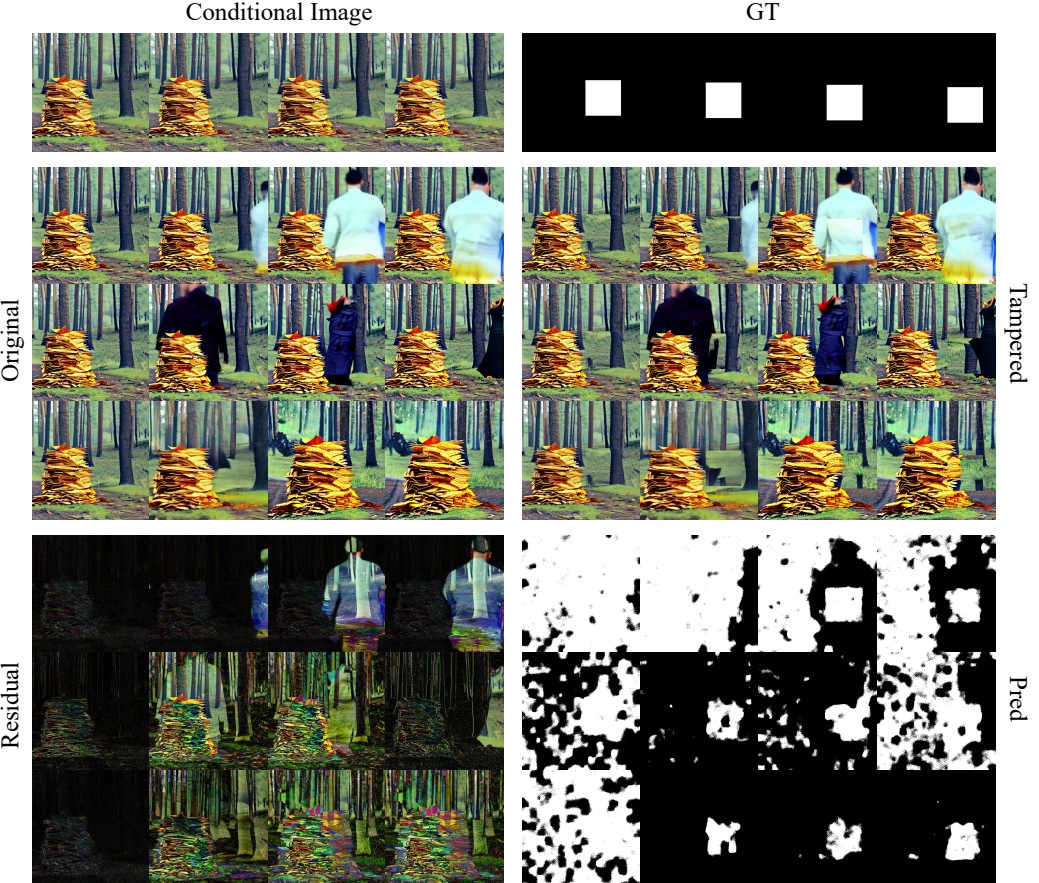

Figure 24: Some videos generated by Stable-Video-Diffusion based on different sampling configurations, along with the corresponding spatial tamper localization results. Each row, from left to right, represents frames 1, 6, 11, and 16 of the generated video. For the original, tampered, residual, and predicted images, the first to third rows correspond to: default settings, Inference Step = 50, and Image Guidance = 2, respectively.

