# OpenReview forum: "VideoShield: Regulating Diffusion-based Video Generation Models via Watermarking"
_ICLR.cc/2025/Conference — ICLR 2025 Poster_

### Official Review · Reviewer_JeCQ · 2024-10-29

**Soundness:** 3
**Presentation:** 3
**Contribution:** 4
**Rating:** 8
**Confidence:** 4

**Summary:**

This article proposes a zero-shot video generation model with active watermarking for video copyright protection and tampering localization.  Based on the DDIM inversion process, the watermark information hidden in the initial Gaussian noise can be detected again from the generation results, and this active watermark is robust against video tampering,  image degradation, and other attacks. Furthermore, for the task of video tampering localization, the authors have designed a detection method that localizes tampering in both temporal (frame position) and spatial  (image information) aspects. Experiments have proven the performance advantages of VideoShield.

**Strengths:**

- VideoShield leverages the unique properties of active watermarking, innovatively combining the model watermarking task with the tampering localization task, demonstrating significant performance advantages.
- This method requires no training and follows a zero-shot paradigm, making it easy to reproduce and validate its performance.
- This method is based on the properties of diffusion and Gaussian distribution, allowing it to be generalized to other diffusion-based models, such as text-to-image models, exhibiting good generalization capabilities.

**Weaknesses:**

- The threshold and hyperparameter settings in the method section are mostly the result of manual searches. The authors may consider exploring how these thresholds could be adaptively searched in the future.

**Questions:**

- In Table 1 and other tables, the metric "Video quality" is mentioned. How is it specifically calculated? The authors mentioned it is the average of a series of metrics; if possible, could you provide the specific values for each metric?
- Currently, the VideoShield method can hide 512 bits. What is the upper limit for watermark capacity?

---

> ### Author Response · Authors · 2024-11-24
>
> # Author Response (Reviewer JeCQ) (1/N)
>
>
> Dear Reviewer WfUe, thank you very much for your careful review of our paper and thoughtful comments. We are encouraged by your positive comments on **the excellent contributions, novelty, and strong generalization capabilities of VideoShield, along with the clear presentation and good soundness of the paper**. We hope the following responses can help clarify potential misunderstandings and alleviate your concerns.
>
> ___
>
> **W1:** The threshold and hyperparameter settings in the method section are mostly the result of manual searches. The authors may consider exploring how these thresholds could be adaptively searched in the future.
>
> **A1:** Thank you for your valuable suggestion.
>
> * We agree that adaptively searching for thresholds could enhance the flexibility and performance of the method. A feasible approach would be to create a small test set of videos and define a target tamper localization performance metric. Using this metric, grid search could be employed to iteratively optimize the relevant thresholds until the desired performance is achieved.
> * However, due to time and resource constraints, we are unable to provide large-scale experimental results for this approach at this time. We appreciate your insightful suggestion and plan to explore this direction in future work.
>
> ___
>
> **Q1:** In Table 1 and other tables, the metric "Video quality" is mentioned. How is it specifically calculated? The authors mentioned it is the average of a series of metrics; if possible, could you provide the specific values for each metric?
>
> **A1:** Thank you for your question. We have provided the calculation details in Appendix D.1 (due to space constraints, we do not list them here) and the specific values for each metric in the Table below. As shown in the Table, **the watermarking method with post-processing causes more significant degradation to the videos in terms of Dynamic Degree and Imaging Quality**. However, for Background Consistency and Motion Smoothness, videos with post-processing sometimes show a slight increase in range. We believe this is due to the watermark embedding introducing some blurring to the video, resulting in better smoothness, which leads to a slight “anomalous” increase in continuity metrics.
>
> **Table:** Specific metric values for video quality assessment.
>
> |         Method         | Subject Consistency | Background Consistency | Motion Smoothness | Dynamic Degree | Imaging Quality |
> |:----------------------:|:-------------------:|:----------------------:|:-----------------:|:--------------:|:---------------:|
> |       ModelScope       |                     |                        |                   |                |                 |
> |         RivaGAN        |        0.922        |          0.951         |       0.960       |      0.540     |      0.648      |
> |          MBRS          |        0.923        |          0.951         |       0.961       |      0.540     |      0.643      |
> |           CIN          |        0.922        |          0.964         |       0.971       |      0.520     |      0.402      |
> |          PIMoG         |        0.920        |          0.960         |       0.972       |      0.510     |      0.405      |
> |         SepMark        |        0.919        |          0.952         |       0.961       |      0.520     |      0.646      |
> |       VideoShield      |        0.923        |          0.954         |       0.961       |      0.545     |      0.648      |
> | Stable-Video-Diffusion |                     |                        |                   |                |                 |
> |         RivaGAN        |        0.938        |          0.966         |       0.957       |      0.710     |      0.609      |
> |          MBRS          |        0.938        |          0.968         |       0.960       |      0.693     |      0.584      |
> |           CIN          |        0.937        |          0.973         |       0.965       |      0.672     |      0.431      |
> |          PIMoG         |        0.936        |          0.971         |       0.965       |      0.672     |      0.427      |
> |         SepMark        |        0.936        |          0.968         |       0.959       |      0.650     |      0.585      |
> |       VideoShield      |        0.939        |          0.968         |       0.957       |      0.710     |      0.607      |

---

> ### Author Response · Authors · 2024-11-24
>
> # Author Response (Reviewer JeCQ) (2/N)
>
> ___
>
> **Q2:** Currently, the VideoShield method can hide 512 bits. What is the upper limit for watermark capacity?
>
> **A2:** Thank you for your question. **The upper limit for watermark capacity in VideoShield is primarily constrained by the video’s resolution (h,w), frame count (f), and the coefficients used ($k_{f}, k_{c}, k_{h}, k_{w}$) in the embedding process as given by the formula: $(f / k_f) * (c / k_c) * (h / k_h) * (w / k_w)$**.
> * Theoretically, the capacity can be increased when using higher resolution, more frames, or smaller coefficients.
> * In practical use, **the upper limit is constrained by the trade-off between watermark capacity and robustness** when the resolution and frame count are fixed. The number of embedded watermark bits can be flexibly adjusted by modifying the coefficient $k_{f}, k_{c}, k_{h}, k_{w}$. However, as the watermark capacity increases, robustness may decline due to the added complexity of embedding more bits.
> * Therefore, while there is no strict upper limit, the capacity should be carefully adjusted to balance robustness and the characteristics of the video.
> * Finally, in practical scenarios, 512 bits is actually quite large and sufficient to distinguish different users.

---

> > ### Comment · Reviewer_JeCQ · 2024-11-25
> >
> > Thank you for your detailed responses to my comments. I appreciate the clarifications provided, and I believe that all the issues I raised have now been effectively addressed.
> >
> > I would like to reaffirm my recognition of the contribution of this work. The novel approach and insights presented in the paper are indeed valuable, and I maintain my support for my rating at this stage.

---

### Official Review · Reviewer_WfUe · 2024-10-31

**Soundness:** 2
**Presentation:** 3
**Contribution:** 2
**Rating:** 5
**Confidence:** 5

**Summary:**

The watermark can actively identify generated content and is currently the primary method used for detection and attribution in diffusion models. However, there has been no watermarking method specifically addressing video generation models based on diffusion models. This paper proposes VIDEOSHIELD to fill this research gap. VIDEOSHIELD introduces a Gaussian Shading watermark embedding method that does not require additional training. In addition to its basic watermarking functionality, VIDEOSHIELD also acts as a fragile watermark, enabling both temporal (across frames) and spatial (within individual frames) tampering localization. Furthermore, it introduces Hierarchical Spatial-Temporal Refinement to enhance accuracy. Experimental results demonstrate that VIDEOSHIELD performs effectively on both T2V and I2V diffusion models.

**Strengths:**

1. This paper introduces a Gaussian Shading watermark embedding method that does not require training, thereby avoiding additional computational overhead.
2. Unlike previous methods that focus solely on spatial tampering localization, VIDEOSHIELD takes into account both temporal and spatial tampering localization during the tampering detection process. The paper introduces Hierarchical Spatial-Temporal Refinement, which enables the generation of more detailed masks for tampering localization, significantly improving the accuracy of the detection.

**Weaknesses:**

1. The innovation presented in this paper is moderate. It extends Gaussian Shading from images to videos and further explores its potential as a fragile watermark.
2. The presentation of the text has significant issues, particularly in Section 3.4, "TAMPER LOCALIZATION." In this section, the extensive use of symbols due to the inclusion of formulas creates obstacles to understanding the paper's content. The frequent reuse of subscripts p and q makes reading quite difficult; the matrix subscript notation is inconsistent, such as in C_(p(q=M(p))), which is recommended to be changed to C_(p,M(p)). Additionally, the comparison bits matrix Cmp could be written as CMP to standardize with other matrix symbols and to avoid confusion with the subscript p. In line 334, the letter m is used both for the variable and for watermark information, which may lead to confusion. After reviewing the main text, I found that in Supplementary Material A3.1, the authors provide a simple example of the process. I strongly recommend including Figure 7 in the main text to aid reader comprehension. Furthermore, there is some overlap between Table 1 and the Datasets section.
3. In my view, the authors have not addressed an important issue regarding the sequence of watermark attribution and tampering localization. The paper does not demonstrate the accuracy of the watermark after spatial tampering. If we assume that a video undergoes spatial tampering and the tampered areas are detected through localization, but the watermark attribution fails, we cannot conclude that a third party maliciously altered the generated video. This raises the question of whether the watermark has become ineffective in such a scenario.

**Questions:**

Address the weakness, especially the novelty issue.

---

> ### Author Response · Authors · 2024-11-24
>
> # Author Response (Reviewer WfUe)
>
>
> Dear Reviewer WfUe, thank you very much for your careful review of our paper and thoughtful comments. We are encouraged by your positive comments on **filling the research gap in watermarking diffusion-based video generation models, reducing computational overhead, and providing a clear presentation of the paper**. We hope the following responses can help clarify potential misunderstandings and alleviate your concerns.
>
> ___
>
> **W1:** The innovation presented in this paper is moderate. It extends Gaussian Shading from images to videos and further explores its potential as a fragile watermark.
>
> **A1:** Thank you for your feedback and we do understand your cocnerns. However, we respectfully argue that our VideoShield is not a straightforward extension of Gaussian Shading (GS) to the video modality. The distinct contributions of our work are outlined as follows:
>
> **1. The Core Role of Template Bits.**
>
> * Template bits form the cornerstone of our framework, **a concept entirely absent in GS**. These bits establish a seamless, training-free connection between watermark embedding, extraction, and tamper localization.
> * Furthermore, template bits are highly versatile, applicable to both images and videos, and adaptable to other modalities. This plug-and-play capability paves the way for multifunctional watermarking solutions, making the framework flexible and widely applicable.
>
> **2. GS as One Implementation Method Among Others.**
> * While GS provides one method to implement template bits, **it is not the only option**. For example, PRC [1] also serves as an alternative implementation.
> * Specifically, PRC employs pseudorandom error-correcting code (PRC) to transform watermark bits into pseudo-random bits that correspond one-to-one with the content. These pseudo-random bits can then be used as template bits. PRC further combines the absolute values of randomly sampled Gaussian noise with the signs of the template bits, embedding the watermark effectively.
>
> **3. Innovations in the Video Modality.**
>
> * Our work introduces **the novel concept of temporal tamper localization**, addressing challenges that traditional methods cannot resolve.
> * Additionally, leveraging the unique characteristics of video data, we propose two innovative techniques, **Hierarchical Spatial-Temporal Refinement** and **Partial Threshold Binarization**, which substantially enhance the robustness of spatial tamper localization.
> * These contributions close critical gaps in tamper localization for video modalities, setting a foundation for more robust and versatile watermarking frameworks.
>
> **References**
> 1. An undetectable watermark for generative image models. arXiv preprint arXiv:2410.07369, 2024.
>
> ___
>
> **W2:** The presentation of the text has significant issues, particularly in Section 3.4, "TAMPER LOCALIZATION."
>
> **A2:** Thank you for your detailed and constructive feedback. We have made the necessary adjustments according to your suggestions:
> * We have revised the notation and standardized the matrix subscripts for better clarity, with the changes highlighted in red in the revised PDF the first time they appear in the main text.
> * Figure 7 has been moved to the main text (it is now Figure 2) to improve reader comprehension.
> * Additionally, we have resolved the overlap between Table 1 and the Datasets section.
>
> We hope these revisions enhance the clarity and readability of the paper.
>
> ___
>
> **W3:** The paper does not demonstrate the accuracy of the watermark after spatial tampering.
>
> **A3:** Thank you for your insightful comment. To address your concern, we have conducted additional experiments that confirm **the watermark remains effectively extractable with near 100% accuracy, even after the video undergoes spatial tampering**. The results of these experiments are provided below. We hope this clarifies the robustness of the watermark in such scenarios.
>
> **Table:** Performance of watermark extraction on the tampered videos.
>
> | Model |  STTN | ProPainter | Crop&Drop |
> |:-----:|:-----:|:----------:|:---------:|
> |   MS  | 0.975 |    0.971   |   0.997   |
> |  SVD  | 0.976 |    0.975   |   0.982   |

---

> > ### Comment · Reviewer_WfUe · 2024-11-27
> > **Reviewer's comment**
> >
> > The authors have addressed some concerns, but the novelty is still not enough for ICLR.

---

### Official Review · Reviewer_g7Rd · 2024-11-02

**Soundness:** 3
**Presentation:** 3
**Contribution:** 3
**Rating:** 8
**Confidence:** 5

**Summary:**

This paper presents a VideoShield, which embeds watermarks directly during video generation. It maps watermark bits to template bits, which are then used to generate watermarked noise during the denoising process. Template bits allow precise detection for potential spatial and temporal modification. Extensive experiments demonstrate its effectiveness.

**Strengths:**

The topic is interesting. Using DDIM Inversion, the video can be reversed to its original watermarked noise. The paper is written and organized well. Experiments are plenty and convincing. The ablation study verified the contribution of each design towards the whole framework.

**Weaknesses:**

No obvious drawbacks are found. However, typesetting needs to be improved, such as keeping the same or similar text size of the table as the text, no overlapping between table and text.

**Questions:**

The framework looks heavy, how about the computational overload?
The relationship with "Gaussian Shading" needs to be further clarified.

---

> ### Author Response · Authors · 2024-11-24
>
> # Author Response (Reviewer g7Rd)
>
>
> Dear Reviewer g7Rd, thank you very much for your careful review of our paper and thoughtful comments. We are encouraged by your positive comments on **the interest of the topic, the clear presentation and organization of the paper, the abundance of convincing experiments, and the good contributions**. We hope the following responses can help clarify potential misunderstandings and alleviate your concerns.
>
> ___
>
> **W1:** Typesetting needs to be improved, such as keeping the same or similar text size of the table as the text, no overlapping between table and text.
>
> **A1:** Thank you for your feedback. We have made the necessary adjustments in the revised PDF to ensure consistent text size in the tables and prevent any overlap with the text. We hope these changes address your concerns.
>
> ___
>
> **Q1:** The framework looks heavy, how about the computational overload? The relationship with "Gaussian Shading" needs to be further clarified.
>
> **A1:** Thank you for your question.
>
> We have provided **the computational overload** in the Table below. From the Table, we have:
> * The primary GPU memory usage and runtime overhead are concentrated in the DDIM Inversion stage.
> * However, as shown in the table for step = 10 and step = 25, **reducing the number of inversion steps can significantly decrease the runtime, with only a slight sacrifice in performance** as shown in Table 11 of the paper.
> * We also provide an analysis of potential speed-up methods for the inversion process in Appendix E.6.
>
> **Table:**
> Computational overhead of VideoShield in terms of GPU memory usage and runtime, evaluated on a single NVIDIA RTX A6000 GPU (49 GB) in FP16 mode. The average runtime of different stages is reported based on 50 generated videos, with the batch size set to 1. “Step” refers to the inversion step, while “Watermark,” “Spatial,” and “Temporal” indicate the average runtime of watermark extraction, spatial tamper localization, and temporal tamper localization, respectively. (s) indicates the runtime in seconds.
>
> |  Model | #Params | Resolution | GPU (GB) | Step=10 (s) | Step=25 (s) | Watermark Extraction (s) | Spatial (s) | Temporal (s) |
> |:------:|:-------:|:----------:|:--------:|:-------:|:-------:|:--------------------:|:-------:|:--------:|
> |   MS   |  1.83B  |     256    |   3.77   |  1.2408 |  3.0617 |        0.0011        |  0.0019 |  0.0004  |
> |   SVD  |  2.25B  |     512    |   5.32   |  4.3214 | 10.2027 |        0.0023        |  0.0019 |  0.0004  |
> |   ZS   |  1.83B  |     256    |   3.77   |  1.2430 |  3.0492 |        0.0011        |  0.0019 |  0.0004  |
> | I2VGen |  2.48B  |     512    |   5.99   |  4.6700 | 11.1526 |        0.0023        |  0.0019 |  0.0004  |
>
> As for **the relationship with "Gaussian Shading"**, in fact, VideoShield is not a straightforward extension of Gaussian Shading (GS) to the video modality. The distinct contributions of our work are outlined as follows:
>
> **1. The Core Role of Template Bits.**
>
> * Template bits form the cornerstone of our framework, **a concept entirely absent in GS**. These bits establish a seamless, training-free connection between watermark embedding, extraction, and tamper localization.
> * Furthermore, template bits are highly versatile, applicable to both images and videos, and adaptable to other modalities. This plug-and-play capability paves the way for multifunctional watermarking solutions, making the framework flexible and widely applicable.
>
> **2. GS as One Implementation Method Among Others.**
> * While GS provides one method to implement template bits, **it is not the only option**. For example, PRC [1] also serves as an alternative implementation.
> * Specifically, PRC employs pseudorandom error-correcting code (PRC) to transform watermark bits into pseudo-random bits that correspond one-to-one with the content. These pseudo-random bits can then be used as template bits. PRC further combines the absolute values of randomly sampled Gaussian noise with the signs of the template bits, embedding the watermark effectively.
>
> **3. Innovations in the Video Modality.**
> * Our work introduces **the novel concept of temporal tamper localization**, addressing challenges that traditional methods cannot resolve.
> * Additionally, leveraging the unique characteristics of video data, we propose two innovative techniques, **Hierarchical Spatial-Temporal Refinement** and **Partial Threshold Binarization**, which substantially enhance the robustness of spatial tamper localization.
> * These contributions close the critical gaps in tamper localization for video modalities, setting a foundation for more robust and versatile watermarking frameworks.
>
> **References**
> 1. An undetectable watermark for generative image models. arXiv preprint arXiv:2410.07369, 2024.

---

> > ### Comment · Reviewer_g7Rd · 2024-11-25
> > **Detailed responses to my concerns**
> >
> > Thank authors for the careful and technical responses, which address my concerns. I keep the ratings.

---

### Official Review · Reviewer_RSGi · 2024-11-02

**Soundness:** 2
**Presentation:** 2
**Contribution:** 2
**Rating:** 3
**Confidence:** 4

**Summary:**

The paper introduces VIDEOSHIELD, a novel watermarking framework for diffusion-based video generation models. By embedding the watermark directly during the generation process, VIDEOSHIELD eliminates the need for additional training, providing a cost-effective solution for safeguarding generated videos. Additionally, the model includes a proactive tamper detection mechanism. Using template bits derived from the watermark, the authors enable both temporal and spatial localization of tampering within the video.

**Strengths:**

- This paper is well-written.
- This paper proposes a novel scheme for proactive video forensics.

**Weaknesses:**

1.	The proposed model may not entirely align with watermarking application scenarios. In its context, verification requires the recipient to compare the watermarked video against the original video's bit template. This requires the verifier to either have access to the original video to generate the bit template using the same encryption method, or to obtain the bit template directly from the video creator. Such requirements differ from standard watermark extraction practices, where verifiers typically can extract watermark information without needing the original video.

2.	The model uses the CHACHA20 stream cipher to generate template bits, requiring the algorithm to use the same key with different nonces for each encryption. Does this mean that a unique random number needs to be set for each video? If so, how does the method handle the storage of a large number of video-to-key mappings? Additionally, it’s unclear whether the primary goal of this approach is to protect the model itself or the generated videos.

3.	The proposed method uses watermarking for proactive tamper detection; however, the baseline experiments in the paper compare it primarily with passive detection methods (e.g., MVSS-Net, HiFi-Net), which may not be entirely appropriate. It would be more suitable to compare the approach with other active tamper detection methods. Below are references to such methods for consideration:
[1] Zhang X, Li R, Yu J, et al. Editguard: Versatile image watermarking for tamper localization and copyright protection[C]//Proceedings of the IEEE/CVF Conference on Computer Vision and Pattern Recognition. 2024: 11964-11974.
[2] Zhou Y, Ying Q, Wang Y, et al. Robust watermarking for video forgery detection with improved imperceptibility and robustness[C]//2022 IEEE 24th International Workshop on Multimedia Signal Processing (MMSP). IEEE, 2022: 1-6.

4.	There are formatting issues: the bottom line of Table 1 overlaps with the text below, and the font size in Table 3 is too small, making it difficult for readers to follow.

**Questions:**

NA

---

> ### Author Response · Authors · 2024-11-24
>
> # Author Response (Reviewer RSGi) (1/N)
>
> Dear Reviewer RSGi, thank you very much for your careful review of our paper and thoughtful comments. We are encouraged by your positive comments on **the novelty of the proposed framework and the well-written structure of the paper**. We hope the following responses can help clarify potential misunderstandings and alleviate your concerns.
>
> ___
>
> **W1&W2:** The proposed model may not entirely align with watermarking application scenarios, where verifiers typically can extract watermark information without needing the original video or the bit template directly from the video creator. Does this mean that a unique random number needs to be set for each video? It’s unclear whether the primary goal of this approach is to protect the model itself or the generated videos.
>
> **A:** Thank you for these insightful comments!
> * These questions pertain to **the scenario of watermark verification, the number of used keys, and the target of protection for the common in-generation watermarking strategy**. It seems there might be some misunderstandings regarding the in-generation watermarking paradigm. To clarify them, we will first outline the general characteristics or use cases of in-generation watermarking methods. Then, we will address the three questions.
> * Existing in-generation watermarking methods, designed for various modalities (e.g., text [1], images [2], proteins [3], etc.), share the following common characteristics:
>     1. **Protection of content generated by closed-source models**.
> 	    * These watermarking methods are designed for closed-source models, where users do not have access to the content generation process. Instead, the model owner (e.g., the platform providing the model service) can intervene in this process to embed watermarks.
> 	    * The ultimate goal is to protect the generated content, with its copyright belonging to the model owner. This approach aligns with the requirements of recent relevant regulations [4][5][6].
>     2. **Watermark verification is performed by the model owner**.
> 	    * As described in characteristic 1, the model owner, being the party responsible for embedding the watermark, has access to all necessary information (e.g., keys, model parameters) required for verification. Therefore, **only the model owner can verify the watermark**, safeguarding their copyright over the generated content.
> 	    * If a legitimate third party wishes to verify the watermark, they must **collaborate with the model owner and obtain the necessary embedding information**.
>     3. **Keys are typically used for watermark embedding, but not for every piece of content**. Most existing in-generation watermarking methods rely on keys during embedding. The model owner generally follows one of two scenarios when setting up keys:
> 	    * For multi-bit watermarks, **a single key related to the model** is used to embed the watermark, with detection focusing solely on the watermark bits themselves for traceability.
> 	    * For zero-bit watermarks, **a unique key is assigned to each user**, allowing the key to serve as a traceable identifier for content-generation users.
>
> Based on these common characteristics, we provide answers to your three questions as follows:
>
> * **Q1:** The proposed model may not entirely align with watermarking application scenarios, where verifiers typically can extract watermark information without needing the original video or the bit template directly from the video creator.
>     * As outlined in characteristic 2, legitimate third verifiers must obtain template bits from the model owner to conduct watermark verification, which is different from the traditional watermarking application scenarios you mentioned.
>
> * **Q2:** Does this mean that a unique random number needs to be set for each video?
>     * As noted in characteristic 3, **it is not necessary to assign a unique key or nonce (random number) to every generated video**. Since we embed multi-bit watermarks, **the model owner only needs to set a single key and nonce (collectively referred to as a “key”)** for one video generation model.
>     * **This does not compromise the diversity of generated videos**. Even if the same key and watermark bits result in identical template bits, the truncated Gaussian sampling (as described in Equation 1 of the paper) produces unique Gaussian noise. Furthermore, the inherent randomness in the sampling process and the different initial Gaussian noise ensure that even videos generated with the same prompt will exhibit diversity.
>
> * **Q3:** It’s unclear whether the primary goal of this approach is to protect the model itself or the generated videos.
>     * As mentioned in characteristic 1, VideoShield is designed to protect video content generated by closed-source models.

---

> ### Author Response · Authors · 2024-11-24
>
> # Author Response (Reviewer RSGi) (2/N)
>
> **References For 1/N**
> 1. Scalable watermarking for identifying large language model outputs. Nature, 2024. https://doi.org/10.1038/s41586-024-08025-4
> 2. Gaussian Shading: Provable Performance-Lossless Image Watermarking for Diffusion Models. CVPR, 2024.
> 3. Enhancing Biosecurity with Watermarked Protein Design[J]. bioRxiv, 2024.
> 4. https://www.reuters.com/technology/artificial-intelligence/openai-supports-california-ai-bill-requiring-watermarking-synthetic-content-2024-08-26
> 5. https://www.europarl.europa.eu/topics/en/article/20230601STO93804/eu-ai-act-first-regulation-on-artificial-intelligence
> 6. https://www.pbs.org/newshour/politics/new-bipartisan-bill-would-require-labeling-of-ai-generated-videos-and-audio
>
> ___
>
> **W3:** It would be more suitable to compare the approach with other proactive tamper detection methods.
>
> **A3:** Thank you for your feedback.
>
> * As the second paper you mentioned is not open-source, we conduct comprehensive experiments to compare VideoShield with EditGuard, the recently open-sourced (after our submission) proactive tamper localization model known for its exceptional performance.
> * Our evaluation covers both videos and images generated by various models. Specifically, we use the default spatial tamper method, Crop&Drop, to create 200 tampered videos for each video generation model and 500 tampered images for each image generation model.
> * As shown in the Table below, **VideoShield outperforms EditGuard on five models**, with the exception of SVD and I2VGen. The relatively lower performance on SVD, in particular, is discussed in Appendix G.3, where we attribute this to the quality of the generated content. **Most importantly, as EditGuard is designed for image modality, it fails for temporal tamper localization.**
>
> **Table:** Comparison of spatial tamper localization performance between VideoShield and EditGuard on videos and images generated by various models. SD stands for Stable Diffusion. The table presents the average values of five metrics: F1, Precision, Recall, AUC, and IoU.
>
> |    Method   |   MS  |  SVD  |   ZS  | I2VGen | SD 1.4 | SD 1.5 | SD 2.1 |
> |:-----------:|:-----:|:-----:|:-----:|:------:|:------:|:------:|:------:|
> |  EditGuard  | 0.890 | 0.886 | 0.880 |  0.882 |  0.885 |  0.885 |  0.885 |
> | VideoShield | 0.906 | 0.761 | 0.900 |  0.867 |  0.938 |  0.939 |  0.928 |
>
> ___
>
> **W4:** There are formatting issues: the bottom line of Table 1 overlaps with the text below, and the font size in Table 3 is too small, making it difficult for readers to follow.
>
> **A4:** Thank you for pointing out the formatting issues. We have addressed these concerns in the revised version of the PDF, where the overlap in Table 1 and the font size issue in Table 3 have been corrected. We hope the revised version resolves these problems, and we appreciate your attention to detail.

---

> ### Author Response · Authors · 2024-12-02
>
> Dear Reviewer,
>
> We hope our responses to your comments have sufficiently addressed your concerns. Currently, your score places the paper below the acceptance threshold, but we believe VideoShield has the potential to meet the standard for publication. If you feel the rebuttal resolves your initial concerns, we kindly ask you to consider adjusting your score accordingly.
>
> Please don’t hesitate to reach out if further clarification is needed. Thank you again for your time and effort!

---

### Official Review · Reviewer_Jpgb · 2024-11-04

**Soundness:** 3
**Presentation:** 3
**Contribution:** 3
**Rating:** 6
**Confidence:** 4

**Summary:**

The paper "VideoShield: Regulating Diffusion-Based Video Generation Models via Watermarking" presents VideoShield, a novel framework for embedding watermarks in diffusion-based video generation models, including both text-to-video (T2V) and image-to-video (I2V) models. The paper addresses the need for content control and authenticity verification in AI-generated video content, focusing on integrating watermarks during the video generation process to prevent quality degradation. Key contributions include In-Generation Watermark Embedding, Tamper Localization for Videos, and Watermark Extraction and Robustness.

**Strengths:**

1. The paper introduces an innovative approach to watermarking in diffusion-based video generation models by embedding watermarks during the generation process, which diverges from traditional post-processing methods. Compared to image watermarking, this is a less explored field. The originality of embedding watermarks during the generation process, as well as the novel dual tampering localization, is a meaningful supplement to this field.
2. The technical methods are rigorous and fully described. This paper uses DDIM inversion to provide a solid foundation for watermark embedding and extraction without affecting video quality. The extensive experimental evaluation of multiple T2V and I2V models strongly demonstrates this method's robustness, flexibility, and effectiveness. The author's detailed analysis of different watermark extraction and localization scenarios further strengthens the contribution of this article to this field.
3. This paper is well-organized and provides a clear understanding of motivation, methods, and experimental setup.
4. VideoShield's training-free and high-fidelity watermarking method provides a reliable and efficient solution for generating watermarks in video models. This paper addresses a highly relevant issue - ensuring the integrity of content in videos generated by artificial intelligence.

**Weaknesses:**

1. Although this article introduces relevant watermarking and tampering localization methods, there is no comparative analysis with other video generation or post-processing watermarking methods. For example, including baselines for time tampering localization or using alternative methods to evaluate the robustness of specific types of distortions.
2. VideoShield is positioned as a non-training and efficient framework, but this paper lacks specific comparisons with baselines regarding time/computational complexity, such as runtime.
3. Although these experiments covered various distortions, they did not explore broader real-world attack scenarios, including testing with denser video compression techniques, color distortion, or frame rate changes, which could further validate the robustness of VideoShield in various real-world environments. In addition, analyzing the performance of VideoShield under adversarial conditions where attackers actively attempt to bypass watermarks could be an interesting solution.
4. The author can further discuss the limitations by listing the performance of VideoShield under different video quality and generation settings. This method seems to perform better on higher-quality video output, but further discussion on factors that may affect watermark integrity, such as video resolution or content complexity, will further demonstrate the effectiveness boundary of VideoShield.
5. This article briefly introduces the adaptability of image watermarking, but can VideoShield adapt to image tampering localization scenarios?

**Questions:**

1. Can the author provide more comparative details on how VideoShield compares to other watermarking frameworks or post-processing methods in terms of tamper localization accuracy and robustness?
2. Can the author share the computational performance of VideoShield, especially regarding the running time of different video resolutions or models?
3. Will the author consider testing the robustness of VideoShield under other types of distortion, such as extreme video compression, frame rate variations, or color adjustments? These additional distortions are common in real-world applications and can enhance confidence in VideoShield's resilience.
4. The results indicate that VideoShield has a certain dependence on video quality. Does the performance of VideoShield still vary due to resolution or model complexity? If so, can the author provide more insights or data on these factors?
5. Can VideoShield adapt to image tampering localization?
6. Is the watermark capacity of 6 512 bits suitable for all video resolutions, or is the number of bits that can be embedded flexible based on the characteristics of the video?
7. Placing visual content in the main text seems better.

---

> ### Author Response · Authors · 2024-11-24
>
> # Author Response (Reviewer Jpgb) (1/N)
>
> Dear Reviewer Jpgb, thank you very much for your careful review of our paper and thoughtful comments. We are encouraged by your positive comments on **the novelty of VideoShield, its significant contributions to the explored field, the rigorous and comprehensive description of the technical methods, the well-organized structure of the paper, the extensive experimental evaluation, and the thorough analysis of various scenarios**. We hope the following responses can help clarify potential misunderstandings and alleviate your concerns.
>
> ___
>
> **W1&Q1:** Can the author provide more comparative details on how VideoShield compares to other watermarking frameworks or post-processing methods in terms of tamper localization accuracy and robustness?
>
> **A1:** Thank you for the valuable feedback. We appreciate the opportunity to clarify the comparative aspects of our work.
> 1. **Watermarking Methods Comparison.**
> 	* Currently, there are **no in-generation video watermarking methods available** for direct comparison. The only available post-processing video watermarking framework is RivaGAN, which has been included as a baseline in Table 1. For the other comparison baselines, we adopted famous and state-of-the-art image watermarking methods, and we leveraged these methods to embed the same watermark bits in each frame of the video, and then conduct relevant tests accordingly.
> 	* The robustness of the watermarking methods is visualized in Figure 7 (in the revised PDF) under different distortions. The specific configurations of different distortions are in *Watermark extraction* section of Appendix D.5:
> 	    * We consider the following distortions: three video distortions—MPEG-4, Frame Average ($N=3$), and Frame Swap ($p=0.25$)—and eight image distortions applied to each video frame: Gaussian Noise ($\sigma=0.1$), Gaussian Blur ($r=4$), Median Blur ($k=7$), Salt and Pepper Noise ($p=0.1$), Random Crop ($0.5$), Random Drop ($0.5$), Resize ($0.3$), and Brightness ($\text{factor}=6$).
>     * Besides, we have provided the additional specific robustness evaluation results in Appendix E.2, **which demonstrate that our VideoShield outperforms all baseline watermarking methods in terms of robustness**.
> 2. **Temporal Tamper Localization Comparison.**
> 	* To the best of our knowledge, we are the first to introduce the task of temporal tamper localization, there is currently no established baseline for comparison in this area.
> 3. **Spatial Tamper Localization Comparison.**
> 	* To the best of our knowledge, there are **no open-source video tamper localization methods available** for direct comparison. At the time of this work, there were also **no open-source proactive image tamper localization methods** suitable for comparison, so we resorted to **open-source passive image tamper localization methods**. We selected several of the most effective methods for comparison, with the results presented in Table 3 of the revision.
> 	* Additionally, in the revised PDF, we have included a comparison with the recently open-sourced state-of-the-art method, EditGuard, which is an proactive image tamper localization approach. **VideoShield outperforms EditGuard on videos or images generated by five out of the seven models**. The detailed comparison results are in the Table below and further analysis can be found in Appendix E.6.
>
> **Table:** Comparison of spatial tamper localization performance between VideoShield and EditGuard on videos and images generated by various models. SD stands for Stable Diffusion. The table presents the average values of five metrics: F1, Precision, Recall, AUC, and IoU.
>
> |    Method   |   MS  |  SVD  |   ZS  | I2VGen | SD 1.4 | SD 1.5 | SD 2.1 |
> |:-----------:|:-----:|:-----:|:-----:|:------:|:------:|:------:|:------:|
> |  EditGuard  | 0.890 | 0.886 | 0.880 |  0.882 |  0.885 |  0.885 |  0.885 |
> | VideoShield | 0.906 | 0.761 | 0.900 |  0.867 |  0.938 |  0.939 |  0.928 |

---

> ### Author Response · Authors · 2024-11-24
>
> # Author Response (Reviewer Jpgb) (2/N)
>
> ___
>
> **W2&Q2:** Can the author share the computational performance of VideoShield, especially regarding the running time of different video resolutions or models?
>
> **A2:** Thank you for your question. We appreciate your interest in the computational performance of VideoShield. We have provided the computational cost in the Table below, including the running time for different video resolutions and models. From the Table, we have:
> * The primary GPU memory usage and runtime overhead are concentrated in the DDIM Inversion stage.
> * However, as shown in the table for step = 10 and step = 25, **reducing the number of inversion steps can significantly decrease the runtime, with only a slight sacrifice in performance** as shown in Table 11 of the paper.
> * We also provide an analysis of potential speed-up methods for the inversion process in Appendix E.7.
>
> We hope this addresses your concern, and we are happy to provide further clarifications if needed.
>
> **Table:**
> Computational overhead of VideoShield in terms of GPU memory usage and runtime, evaluated on a single NVIDIA RTX A6000 GPU (49 GB) in FP16 mode. The average runtime of different stages is reported based on 50 generated videos, with the batch size set to 1. “Step” refers to the inversion step, while “Watermark,” “Spatial,” and “Temporal” indicate the average runtime of watermark extraction, spatial tamper localization, and temporal tamper localization, respectively. (s) indicates the runtime in seconds.
>
> |  Model | #Params | Resolution | GPU (GB) | Step=10 (s) | Step=25 (s) | Watermark Extraction (s) | Spatial (s) | Temporal (s) |
> |:------:|:-------:|:----------:|:--------:|:-------:|:-------:|:--------------------:|:-------:|:--------:|
> |   MS   |  1.83B  |     256    |   3.77   |  1.2408 |  3.0617 |        0.0011        |  0.0019 |  0.0004  |
> |   SVD  |  2.25B  |     512    |   5.32   |  4.3214 | 10.2027 |        0.0023        |  0.0019 |  0.0004  |
> |   ZS   |  1.83B  |     256    |   3.77   |  1.2430 |  3.0492 |        0.0011        |  0.0019 |  0.0004  |
> | I2VGen |  2.48B  |     512    |   5.99   |  4.6700 | 11.1526 |        0.0023        |  0.0019 |  0.0004  |

---

> ### Author Response · Authors · 2024-11-24
>
> # Author Response (Reviewer Jpgb) (3/N)
>
> ___
>
> **W3&Q3:** Will the author consider testing the robustness of VideoShield under other types of distortion, such as extreme video compression, frame rate variations, or color adjustments?
>
> **A3:** Thank you for your insightful question.
>
> 1. **For frame rate variations**, there are two scenarios:
> 	* **Altering the playback duration without changing the number of frames**. In this scenario, VideoShield’s performance remains unaffected as the tested frames are not altered.
> 	* **Modifying the number of frames through frame insertion or deletion while keeping the playback duration unchanged**. This essentially corresponds to temporal tampering, and we have already included the localization accuracy for Frame Insert and Drop in Table 2 in the revised PDF.
> 2. **For video compression**, the results are shown in the Table below (CRF). The typical CRF (Constant Rate Factor) range is 18–28, where 18 corresponds to higher video quality and 28 to lower quality.
> 	* The results show that **only under extreme compression conditions (CRF > 28) do the watermark extraction and localization performance for both MS and SVD start to degrade significantly**. However, at this point, the video quality is already severely degraded, making the performance drop acceptable.
> 	* Besides, **adjusting the threshold substantially improves spatial localization performance**.
> 3. **For color adjustments**, the results are shown in the Table below (Hue). VideoShield shows **strong robustness, maintaining high performance in both watermark extraction and tamper localization**.
>
> Overall, VideoShield demonstrates good robustness, making it suitable for most real-world scenarios. We hope this addresses your concerns and are happy to provide further clarifications if needed.
>
> **Table:** Performance under more spatial distortions. CRF represents different compression factors of H.264. As discussed in Section 3.5, 'w/o' refers to using $t_{wm}$ and $t_{orig}$ derived from the clean watermarked videos, while 'w/' indicates that various distortions are introduced during the distribution testing, leading to adjustments in $t_{wm}$ and $t_{orig}$. Hue=0.5 is the max factor we can configure.
>
> | Dimension |  Model  | Clean | CRF=18 | CRF=23 | CRF=28 | CRF=33 | CRF=38 | Hue=0.5 |
> |:---------:|:-------:|:-----:|:------:|:------:|:------:|:------:|:------:|:-------:|
> | Watermark |    MS   | 1.000 |  0.999 |  0.999 |  0.997 |  0.978 |  0.907 |  1.000  |
> |           |   SVD   | 0.999 |  0.988 |  0.980 |  0.961 |  0.933 |  0.888 |  0.993  |
> |  Spatial  |  MS w/o | 0.906 |  0.861 |  0.826 |  0.746 |  0.639 |  0.555 |  0.842  |
> |           |  MS w/  | 0.806 |  0.814 |  0.812 |  0.788 |  0.720 |  0.627 |  0.815  |
> |           | SVD w/o | 0.761 |  0.667 |  0.635 |  0.607 |  0.577 |  0.547 |  0.684  |
> |           |  SVD w/ | 0.708 |  0.718 |  0.699 |  0.664 |  0.639 |  0.605 |  0.724  |

---

> ### Author Response · Authors · 2024-11-24
>
> # Author Response (Reviewer Jpgb) (4/N)
>
> ___
>
> **W4&Q4:** Does the performance of VideoShield still vary due to resolution or model complexity? If so, can the author provide more insights or data on these factors?
>
> **A4:** Thank you for your thoughtful question.
>
> 1. **For resolution**, since ZeroScope is trained on videos of various resolutions, we choose it to generate videos at different resolutions for this analysis. As shown in the Table below, **VideoShield’s spatial tamper localization performance improves as the resolution increases**. We believe this is because **higher video resolutions capture more details and provide better quality, allowing the template bits to be better integrated into the generated video**. This, in turn, enhances the inversion accuracy, leading to improved spatial tamper localization performance.
> 2. **For model complexity**, a common measure is the number of parameters. To explore this, one would need models of the same type (e.g., T2V) with similar architectures but different parameter counts. The T2V models used in our work have 1.83B parameters, while the I2V models are around 2.3B. **Currently, we are unable to find any open-source models that meet this specific requirement**. Therefore, we have not been able to explore this relationship in our current study. However, this is indeed an interesting direction for future work, and we appreciate your valuable suggestion!
>
> **Table:** Performance on videos generated by ZeroScope with different resolutions.
>
> | Resolution |  256  |  320  |  384  |  448  |
> |:----------:|:-----:|:-----:|:-----:|:-----:|
> |  Watermark | 1.000 | 1.000 | 1.000 | 1.000 |
> |  Temporal  | 1.000 | 1.000 | 1.000 | 1.000 |
> |   Spatial  | 0.890 | 0.908 | 0.931 | 0.938 |
>
> ___
>
> **W5&Q5:** Can VideoShield adapt to image tampering localization?
>
> **A5:** Thank you for your question. We appreciate your interest in extending VideoShield to image tampering localization.
> * As discussed in Section 3.4, under “*Spatial tamper localization on images generated by T2I models*”, we have already presented relevant results, which you might have overlooked.
> * Additionally, we have provided a comparison in Appendix E.6 with EditGuard, a state-of-the-art open-source method for proactive image tamper localization. **Our method demonstrates superior performance, further confirming its versatility and effectiveness**. We hope this addresses your concern and welcome any further questions.
>
> For your convenience, we provide the results in the Table below.
>
> **Table:** Comparison of image spatial tamper localization performance between VideoShield and EditGuard on images generated by various models. SD stands for Stable Diffusion. The table presents the average values of five metrics: F1, Precision, Recall, AUC, and IoU.
>
> |    Method   | SD 1.4 | SD 1.5 | SD 2.1 |
> |:-----------:|:------:|:------:|:------:|
> |  EditGuard  |  0.885 |  0.885 |  0.885 |
> | VideoShield |  0.938 |  0.939 |  0.928 |
>
> ___
>
> **Q6:** Is the watermark capacity of 6 512 bits suitable for all video resolutions, or is the number of bits that can be embedded flexible based on the characteristics of the video?
>
> **A6:** **The number of embedded watermark bits can be flexibly adjusted based on video characteristics.** The number of embedded bits is determined by $(f / k_f) * (c / k_c) * (h / k_h) * (w / k_w)$, leading to two possible scenarios:
>
> 1.	**If the coefficient $k_{f}, k_{c}, k_{h}, k_{w}$ is fixed, the number of embedded watermark bits increases with the video resolution and the number of frames.**
> 2.	**If the video resolution and frame count are fixed, the number of embedded watermark bits can be flexibly adjusted by modifying the coefficient $k_{f}, k_{c}, k_{h}, k_{w}$.** However, this introduces a trade-off between watermark extraction robustness and watermark capacity. A larger coefficient means each watermark bit is repeated more times before being embedded into the Gaussian noise, making it determined by more video pixels. This increases fault tolerance, thereby enhancing robustness.
>
> ___
>
> **Q7:** Placing visual content in the main text seems better.
>
> **A7:** Thank you for the suggestion. We agree that placing visual content in the main text can enhance clarity.
>
> * To address this, we have included the comparison of spatial tampering localization results between VideoShield and baselines (**Figure 5**), as well as the illustration of the temporal tamper localization process (**Figure 2**), in the main text, as we consider these two figures particularly important.
> * Due to space constraints, we were unable to include additional visual content in the main text but have provided comprehensive supplementary visualizations in the Appendix.
>
> We hope this strikes a balance and welcome further suggestions.

---

> ### Comment · Reviewer_Jpgb · 2024-11-27
>
> I appreciate and acknowledge the author's emphasis on this work and the effort put into this response, and  I will maintain my current rating.

---

### Official Review · Reviewer_YUmj · 2024-11-04

**Soundness:** 2
**Presentation:** 1
**Contribution:** 2
**Rating:** 6
**Confidence:** 3

**Summary:**

The paper addresses the need for the control of integrity and misuse of AI Generated Content. The usual approach is to use watermarks for video domain, but they are underdeveloped and have a post-processing manner, which results in video quality degradation. The authors propose a novel way of watermarking images while generating it (VideoShield). Their method is training-free and works with diffusion-based video models. Authors extract the watermark from the images using DDIM Inversion.

**Strengths:**

1) The proposed method does not require additional training, which could simplify integration with existing diffusion models.
2) The framework’s ability to detect tampering both within frames and across the sequence

**Weaknesses:**

1) Notation T_temp for a threshold in tampering is a bit confusing, considering T_p is a tensor.
2) No introduction for chacha20
3) No formula for calculating video quality is given
4) Figure 2 provides only abbreviations in the legend
5) Over all paper, formulas and tables seem to have decreased font. Moreover, some tables overlap the text (e.g. Table1). This may be a reason for a possible desk rejection.

**Questions:**

1) Results from Table 1 shows that the proposed method is only marginally better than RivaGAN w/o real images conditioning. Did you check RivaGAN with the same setup as the last row?
2) Why do you compare only to SVD in Table 2?
3) The authors appeal to decreased visual quality of videos generated with the existing watermarking methods, but did not organise subjective study to show that their method outperforms the others. Video quality metrics are known to have limited capabilities to estimate AIGC. Subjective comparison is required in this work, can you provide one?

---

> ### Author Response · Authors · 2024-11-24
>
> # Author Response (Reviewer YUmj) (1/N)
>
>
> Dear Reviewer YUmj, thank you very much for your careful review of our paper and thoughtful comments. We are encouraged by your positive comments on **the novelty and practicality of VideoShield**. We hope the following responses can help clarify potential misunderstandings and alleviate your concerns.
>
> ___
>
> **W1:** Notation $T_{temp}$ for a threshold in tampering is a bit confusing, considering $T_{p}$ is a tensor.
>
> **A1:** Thank you for this suggestion!
> - We acknowledge that the current notation $T_{temp}$ may cause confusion, especially since $T_p$ is a tensor.
> - To address this, we have revised $T_{temp}$ to $t_{temp}$, along with changing $T_{wm}$ to $t_{wm}$ and $T_{orig}$ to $t_{orig}$ in the revised PDF.
>
> These changes are highlighted in red the first time they appear, for your convenience, to clearly reflect the scalar nature of these terms.
> ___
>
> **W2:** No introduction for chacha20
>
> **A2:** Thank you for pointing it out! We have added the introduction below for ChaCha20 in Section 2.3 of the revised PDF and highlighted the changes in blue for your convenience:
>
> * ChaCha20 is a stream cipher that takes a 256-bit key, a 96-bit nonce, and plaintext as input to produce a pseudo-random ciphertext as output. The algorithm generates a keystream that is XORed with the plaintext for encryption.
>
> We hope this addition clarifies the context.
>
> ___
>
> **W3:** No formula for calculating video quality is given
>
> **A3:** Thank you for pointing this out! Due to space limitations in the initial submission, we did not include the formula for calculating video quality.
>
> * We have now added the formulas in Appendix D.1 of the revised PDF for your reference.
> * We also present the formulas below for your convenience:
>     * **Subject Consistency**. Subject Consistency is acquired by calculating the DINO [2] feature similarity across frames:
> \begin{equation}
>     S_{\text{subject}} = \frac{1}{T - 1} \sum_{t=2}^{T} \frac{1}{2} \left( \langle d_1 \cdot d_t \rangle + \langle d_{t-1} \cdot d_t \rangle \right),
> \end{equation}
> where $d_{i}$ is the DINO image feature of the $i^{th}$ frame, normalized to unit length, and $\langle \cdot \rangle$ is the dot product operation for calculating cosine similarity.
>     * **Background Consistency**. Background Consistency evaluates the temporal consistency of the background scenes by calculating CLIP [3] feature similarity across frames:
> \begin{equation}
>     S_{\text{background}} = \frac{1}{T - 1} \sum_{t=2}^{T} \frac{1}{2} \left( \langle c_1 \cdot c_t \rangle + \langle c_{t-1} \cdot c_t \rangle \right),
> \end{equation}
> where $c_{i}$ represents the CLIP image feature of the $i^{th}$ frame, normalized to unit length.
>     * **Motion Smoothness**. Motion Smoothness is evaluated by the frame-by-frame motion prior to video frame interpolation models [4]. Specifically, given a generated video consisting of frames $[f_0, f_1, f_2, f_3, f_4, ..., f_{2n-2}, f_{2n-1}, f_{2n}]$, the odd-number frames are manually dropped to obtain a lower-frame-rate video $[f_0, f_2, f_4, ..., f_{2n-2}, f_{2n}]$, and video frame interpolation [4] is used to infer the dropped frames [ f̂\_1, f̂\_3, ..., f̂\_(2n-1) ]. Then the Mean Absolute Error (MAE) between the reconstructed frames and the original dropped frames is computed and normalized into $[0, 1]$, with a larger number implying smoother motion.
>     * **Dynamic Degree**. Dynamic Degree is designed to assess the extent to which models tend to generate non-static videos. RAFT [5] is used to estimate optical flow strengths between consecutive frames of a generated video. Then the average of the largest 5\% optical flows (considering the movement of small objects in the video) is taken as the basis to determine whether the video is static. The final dynamic degree score is calculated by measuring the proportion of non-static videos generated by the model.
>     * **Imaging Quality**. Imaging Quality is measured by the MUSIQ [6] image quality predictor trained on the SPAQ [7] dataset, which is capable of handling variable-sized aspect ratios and resolutions. The frame-wise score is linearly normalized to [0, 1] by dividing 100, and the final score is then calculated by averaging the frame-wise scores across the entire video sequence.
>
> **References**
> 1. Vbench: Comprehensive benchmark suite for video generative models. CVPR, 2024.
> 2. Emerging properties in self-supervised vision transformers. ICCV, 2021.
> 3. Learning transferable visual models from natural language supervision. ICML, 2021.
> 4. Amt: All-pairs multi-field transforms for efficient frame interpolation. CVPR, 2023.
> 5. Raft: Recurrent all-pairs field transforms for optical flow. ECCV, 2020.
> 6. Musiq: Multi-scale image quality transformer. ICCV, 2021.
> 7. Perceptual quality assessment of smartphone photography. CVPR, 2020.

---

> ### Author Response · Authors · 2024-11-24
>
> # Author Response (Reviewer YUmj) (2/N)
>
> ___
>
> **W4:** Figure 2 provides only abbreviations in the legend
>
> **A4:** Thank you for pointing this out!
> * In our paper,
>     * GA stands for Gather and Average.
>     * PTB stands for Partial Threshold Binarization.
>     * R stands for Repeat.
> * In response, we have revised the legend of Figure 2 (Figure 3 in the revision) in the revised PDF by replacing the abbreviations with their full terms for better clarity.
> ___
>
> **W5:** Over all paper, formulas and tables seem to have decreased font. Moreover, some tables overlap the text (e.g. Table1). This may be a reason for a possible desk rejection.
>
> **A5:** Thank you for pointing this out! In response, we have adjusted the font size of the formulas and tables in the revised PDF, and have corrected the space between the tables and the text to ensure better readability.
>
> ___
>
> **Q1:** Results from Table 1 shows that the proposed method is only marginally better than RivaGAN w/o real images conditioning. Did you check RivaGAN with the same setup as the last row?
>
> **A1:** Thank you for pointing this out and we do understand your concern.
>
> - In the same setup (w/ real images), RivaGAN achieves a Video Quality score of 0.865, which is on par with our method, as shown in the Table below.
> - In fact, RivaGAN is a post-processing video watermarking method, which results in relatively high fidelity. However, besides Video Quality, watermark embedding capacity and robustness are two other important metrics to measure watermarking's performance. As shown in the Table below, **its watermark embedding capacity (Bit Length) and robustness (Bit Accuracy (adv)) are inferior to those of VideoShield**.
>
> **Table:** Comparison of RivaGAN and our VideoShield on videos generated by SVD conditioned on real images. “Bit Accuracy (adv)” refers to the average bit accuracy under various distortions, indicating robustness.
>
> |    Method   | Bit Length | Video Quality | Bit Accuracy | Bit Accuracy (adv) |
> |:-----------:|:----------:|:-------------:|:------------:|:----------:|
> |   RivaGAN   |     32     |     0.865     |     0.989    |    0.890   |
> | VideoShield |     512    |     0.865     |     0.999    |    0.959   |
>
> ___
>
> **Q2:** Why do you compare only to SVD in Table 2?
>
> **A2:** Thank you for this insightful question!
>
> * Here might be some potential misunderstandings
>     * **SVD stands for 'Stable Video Diffusion'** (instead of 'Singular Value Decomposition').
>     * **SVD is not a baseline for comparison**. It is representative of the Image-to-Video (I2V) model used to generate the videos, which are used to evaluate the temporal tamper localization.
> * Table 2 presents the results for temporal tamper localization. Specifically, we include the performance on videos generated by our two default video generation models: the Text-to-Video (T2V) model, MS, and the Image-to-Video (I2V) model, SVD. **Our VideoShield achieves temporal tamper localization accuracies of 1.000 and 0.936 on videos generated by MS and SVD, respectively, demonstrating its outstanding performance.**
> * To the best of our knoledge, we are the first to evaluate temporal teamper localization on generated video, so **there are no existing baselines for direct comparison**. We are very willing to incorporate more comparison for suitable methods if you can kindly provide more details.
> * To ensure clarity, we have presented the results of VideoShield (including watermarking and tamper localization) applied to videos generated by SVD conditioned on real images in a separate Table 4 in the revised PDF.
>
> ___
>
> **Q3:** Subjective comparison is required in this work, can you provide one?
>
> **A3:** Thank you for this constructive suggestion!
>
> * **We further conduct a user study to perform a subjective comparison of video quality**. We randomly select 10 videos from each of the 200 videos generated by MS and SVD. After watermarking these videos by different methods, we distribute the watermarked videos to 24 distinct users with the instruction: “Please choose the video that you consider to have the highest quality, based on Subject Consistency, Background Consistency, Motion Smoothness, Dynamic Degree, and Imaging Quality.”
> * The results are shown in the table below. As observed, VideoShield receives the most votes for both MS and SVD, i.e., **VideoShield achieves the best video quality**, followed closely by RivaGAN with a slight gap, while MBRS also garners relatively high votes. Conversely, CIN, PIMoG, and SepMark receive very few votes, indicating more significant visible degradation in video quality.
>
> **Table:** Subjective comparison results.
>
> | Method | RivaGAN | MBRS | CIN | PIMoG | SepMark | VideoShield |
> |:------:|:-------:|:----:|:---:|:-----:|:-------:|:-----------:|
> |   MS   |    78   |  59  |  3  |   2   |    6    |      92     |
> |   SVD  |    82   |  51  |  1  |   0   |    8    |      98     |

---

> > ### Comment · Reviewer_YUmj · 2024-11-26
> > **Reviewer's response to authors**
> >
> > Thank you for your response. I value your detailed clarifications. I have another question: will you provide the code and video examples for your work?

---

> > > ### Author Response · Authors · 2024-11-27
> > >
> > > Thank you for your follow-up question and for your appreciation of our clarifications. We have uploaded the code and video examples to the supplementary materials, which you can download to have a review locally. We have tried our best to ensure the code are readily accessible and reproducible. Additionally, we plan to upload these materials to a GitHub repository shortly, the open-source link will be also provided once it is available.
> > >
> > > If you encounter any issues with the code or have any suggestions, we warmly welcome discussions and are happy to address any concerns. We hope this will provide additional clarity and support for our contributions.
> > >
> > > We would also greatly appreciate your feedback on whether these materials address any remaining questions or concerns. Should there be no further issues, we would be sincerely grateful if you could consider raising the rating of our submission.

---

> > > ### Author Response · Authors · 2024-11-30
> > >
> > > The code and video examples are now available on an anonymous GitHub repository: https://anonymous.4open.science/r/VideoShield-BB34. Please feel free to review them, and let us know if you have any questions or feedback. We appreciate your consideration of our work!

---

### Author Response · Authors · 2024-11-24

# General Response (All Reviewers)


Dear reviewers,

We would like to express our sincere gratitude for your thorough and constructive feedback. Your valuable insights significantly contribute to the improvement of our work. We deeply appreciate the time and effort you dedicate to reviewing our paper.

Based on your suggestions, we make the following revisions:

- We move the "Background" section to Appendix A to reduce the overall length and improve the global layout.
- We adjust the fonts for all formulas and tables, as well as the space between tables and text. (YUmj, RSGi, g7Rd, WfUe)
- We move two important figures from the appendix to the main text: Figure 2 and Figure 5. (Jpgb, WfUe)
- We expand the legend of Figure 3, replacing abbreviations with full terms. (YUmj)
- In Section 2.3, we introduce ChaCha20 (highlighted in blue). (YUmj)
- In Section 2, we make some adjustments to the notations, highlighting their first appearances in red, to avoid confusion and enhance the readability of the paper. (YUmj, WfUe)
- In Section 3.2, we place the SVD results conditioned on real images in a separate Table 4, to avoid any confusion. (YUmj)
- In Appendix B, we further elaborate on the relationship between VideoShield and Gaussian Shading. (g7Rd, WfUe)
- In Appendix D.1, we provide the detailed calculation methods for the video quality metrics. (YUmj, JeCQ)
- In Appendix E.1, we include the baseline comparison with specific video quality evaluation metrics, along with subjective evaluation results for video quality comparison. (YUmj, JeCQ)
- In Appendix E.2, we provide the baseline comparison results for watermark robustness evaluation. (Jpgb)
- In Appendix E.4, we assess the robustness of watermark extraction against various types of spatial tamper. (WfUe)
- In Appendix E.5, we present and analyze the robustness of VideoShield against distortions in more real-world scenarios. (Jpgb)
- In Appendix E.6, we compare and analyze VideoShield with the recently released open-source active image tamper localization method, EditGuard. (RSGi)
- In Appendix E.7, we present and analyze the computational overhead of VideoShield. (Jpgb, g7Rd)
- In Appendix F, we include the results and analysis of VideoShield's performance on videos generated at different resolutions (highlighted in blue). (Jpgb)

We hope these revisions address your concerns and improve the clarity and quality of the paper. Once again, thank you for your valuable feedback and support.

---

### Comment · Area_Chair_yJCE · 2024-11-25

Hi Reviewers,

We are approaching the deadline for author-reviewer discussion phase. Authors has already provided their rebuttal. In case you haven't checked them, please look at them ASAP. Thanks a million for your help!

---

### Meta-Review · Area_Chair_yJCE · 2024-12-20

**Metareview:**

This paper works on watermarking for diffusion-based video generation models. Authors proposed VideoShield for generative video watermarking. VideoShield embeds watermarks during video generation avoiding the need for additional training. Authors proposed a tamper localization feature to detect changes both temporally and spatially. The extraction is done with DDIM inversion. Experimental results show the effectiveness of the proposal methods.

This paper was reviewed by 6 reviewers and got mixed scores as two 8, two 6, one 5, one 3.

Strength and weaknesses given by reviewers before rebuttals are as follows (notes that different reviewers has different perspectives of the paper, so conflicts in strength and weaknesses might happen):

Strength: 1) proposed method doesn't require additional training which is novel; 2) proposed methods could detect tampering both within frames and across the sequence; 3) paper is well written; 4) experimental results are convincing.

Weaknesses: 1) lack of comparative analysis with other video generation or post-processing watermarking methods; 2) lacks comparisons with baselines on time/computational complexity; 3) did not explore broader real-world attack; 4) proposed method may not align with watermarking application scenarios; 5) comparison with other active tamper detection methods; 6) formatting issues; 7) innovation is moderate; 8)  does not demonstrate the accuracy of the watermark after spatial tampering. 9) The threshold and hyperparameter settings in the method section are mostly the result of manual searches.

During rebuttal:
Reviewer YUmj (rating 6) thought their concerns are addressed by authors and increased rating.

Reviewer Jpgb (rating 6), Reviewer g7Rd (rating 8) and Reviewer JeCQ (rating 8) suggested authors addressed their concerns and maintained rating.

Reviewer RSGi (rating 3) didn't provide feedback in the rebuttal.

Reviewer WfUe (rating 5) suggested authors have addressed some concerns, but thought the novelty is still not enough for ICLR.

Several reviewers endorsed the novelty of this paper and were positive about the acceptance of this paper. Also AC checked authors addressed Reviewer RSGi (rating 3)'s concerns, so AC decided to accept this paper.

**Additional Comments On Reviewer Discussion:**

During rebuttal:
Reviewer YUmj (rating 6) thought their concerns are addressed by authors and increased rating.

Reviewer Jpgb (rating 6), Reviewer g7Rd (rating 8) and Reviewer JeCQ (rating 8) suggested authors addressed their concerns and maintained rating.

Reviewer RSGi (rating 3) didn't provide feedback in the rebuttal.

Reviewer WfUe (rating 5) suggested authors have addressed some concerns, but thought the novelty is still not enough for ICLR.

Several reviewers endorsed the novelty of this paper and were positive about the acceptance of this paper. Also AC checked authors addressed Reviewer RSGi (rating 3)'s concerns, so AC decided to accept this paper.

---

### Decision · Program_Chairs · 2025-01-22

Accept (Poster)